# On the Limit to the Accuracy of Regional-Scale Air Quality Models

S. Trivikrama Rao[a,b], Huiying Luo[b], Marina Astitha[b], Christian Hogrefe[c], Valerie Garcia[c], Rohit Mathur[c]

[a]Department of Marine, Earth, and Atmospheric Sciences, North Carolina State University, Raleigh, NC
[b]Department of Civil and Environmental Engineering, University of Connecticut, Storrs, CT
[c]Center for Environmental Measurement & Modeling, U.S. Environmental Protection Agency, Research Triangle Park, NC

*Correspondence to*: S. Trivikrama Rao (strao@ncsu.edu)

**Abstract.** Regional-scale air pollution models are routinely being used world-wide for research, forecasting air quality, and
regulatory purposes. It is well recognized that there are both reducible (systematic) and irreducible (unsystematic) errors in the
meteorology-atmospheric chemistry modeling systems. The inherent (random) uncertainty stems from our inability to properly
characterize stochastic variations in atmospheric dynamics and chemistry, and from the incommensurability associated with
comparisons of the volume-averaged model estimates with point measurements. Because these stochastic variations are not
being explicitly simulated in the current generation of regional-scale meteorology-air quality models, one should expect to
find differences between the model estimates and corresponding observations. This paper presents an observation-based
methodology to determine the expected errors from current generation regional air quality models even when the model design,
physics, chemistry, and numerical analysis, as well as its input data, were "perfect". To this end, the short-term synoptic-scale
fluctuations embedded in the daily maximum 8-hr ozone time series are separated from the longer-term forcing using a simple
recursive moving average filter. The inherent uncertainty attributable to the stochastic nature of the atmosphere is determined
based on 30+ years of historical ozone time series data measured at various monitoring sites in the contiguous United States.
The results reveal that the expected root mean square error at the median and 95th percentile is about 2 ppb and 5 ppb,
respectively, even for "perfect" air quality models driven with "perfect" input data. Quantitative estimation of the limit to the
model's accuracy will help in objectively assessing the current state-of-the-science in regional air pollution models, measuring
progress in their evolution, and providing meaningful and firm targets for improvements in their accuracy relative to ambient
measurements.

## 1 Introduction

Confidence in model estimates of pollutant distributions is established through direct comparisons of modeled concentrations
with corresponding observations made at discrete locations for retrospective cases. Pinder et al. (2008) discussed the reducible
(i.e., structural and parametric) uncertainties that are attributable to the errors in model input data (e.g., meteorology, emissions,
initial and boundary conditions) as well as our incomplete or inadequate understanding of the relevant atmospheric processes
(e.g. chemical transformation, planetary boundary layer evolution, transport and dispersion, modeling domain, grid resolution,

deposition, rain, clouds). Inherent or irreducible (random or unsystematic) uncertainties stem from our inability to properly characterize the stochastic nature of the atmosphere (Wilmott, 1981, 1985: Fox, 1984; Rao et al., 1985; Dennis et al., 2010; Rao et al., 2011) and from the incommensurability associated with comparing the volume-averaged model estimates with point measurements (e.g., McNair et al., 1996; Swall and Foley, 2009). Also, without completely knowing the 3-dimensional initial

physical and chemical state of the atmosphere, its future state cannot be simulated accurately (Lamb, 1984; Lamb and Hati, 1987; Lewellen and Sykes, 1989; Pielke, 1998; Gilliam et al., 2015). Given the presence of the irreducible uncertainties, precise replication of observed concentrations or their changes by the models cannot be expected (Dennis et al., 2010; Rao et al., 2011; Porter et al., 2015; Astitha, 2017)).

Whereas an air quality model's prediction represents some time/space-averaged concentrations, an observation at any given time at a monitoring location reflects an individual event or specific realization out of a population that will almost always differ from the model estimate even if the model and its input data were perfect (Rao et al., 1985). Consequently, comparisons of modeled and observed concentrations paired in space and time indicate biases and errors in simulating absolute levels of pollutant concentrations at individual monitoring sites (Porter et al., 2015). The scientific discussion on modeling uncertainty

goes back more than three decades with the current practice including data assimilation, ensemble modeling, and model performance evaluation (e.g., Fox, 1981, 1984; Lamb, 1984; Demerjian, 1985; Oreskes et al., 1994; Pielke, 1998; Lewellen and Sykes, 1989; Lee et al., 1997; Carmichael et al., 2008; Hogrefe et al., 2001a, 2001b; Biswas and Rao, 2001; Grell and Baklanov, 2011; Gilliam et al., 2006; Herwehe et al., 2011; Baklanov et al., 2014; Bocquet et al., 2015; Solazzo and Galmarini, 2015; Ying and Zhang, 2018; McNider and Pour-Biazar, 2020; Stockwell et al., 2020). While ever-improving process

knowledge and increasing computational power will continue to help reduce the structural and parametric uncertainties in air quality models, the inherent uncertainty associated with our inability to properly characterize the stochastic nature of the atmosphere will always result in some mismatch between the model results and measurements; this could lead to speculation on the inferred accuracy of the future states simulated by the regional-scale air quality models (Dennis et al., 2011; Rao et al., 2011; Porter et al., 2015; Astitha et al., 2017; Luo et al., 2019).

The sensitivity of model results to meteorology, chemical mechanisms, and emissions has been examined in numerous studies (e.g., Vautard et al., 2012; Sarwar et al., 2013; Pierce et al., 2010; Napelenok et al., 2011; Kang et al.,2013). Herwehe et al. (2011) attributed the differences in ground-level ozone predictions between WRF-Chem and WRF-CMAQ models to the way meteorology and chemistry interactions are handled within these two modeling systems. Thomas et al. (2019) examined the ozone predictions in the Mid-Atlantic region of the United States during June 2016 through a series of simulations with WRF-

Chem, focusing on the sensitivity to the meteorological initial/boundary conditions (IC/BCs), emissions inventory (EI), and planetary boundary layer (PBL) scheme. Ying and Zhang (2018) discussed the use of satellite-based observations for improving the predictability of multiscale tropical weather and equatorial waves. Ensemble modeling is being advocated for quantifying the uncertainty in model predictions; however, the spread in the model estimates for the variable of interest reflects the impact of our incomplete or inadequate knowledge of the physical and chemical processes (i.e., the reducible errors

stemming from structural and parametric uncertainty) occurring in the atmosphere (Solazzo et al., 2015; Thomas, et al., 2019; Stockwell et al., 2020). McNider and Pour-Biazar (2020) reviewed the many issues in predicting the prevailing meteorology for regional air quality simulations and indicated that errors in the specification of the physical atmosphere such as temperature, winds, and mixing heights can affect the air quality predictions. Stockwell et al. (2020) discussed the problems relating to the atmospheric chemical mechanisms currently being used for simulating air quality. The current generation of regional models consider only the mean values of a meteorological variable for a given timescale and the average rate constant derived from gas chamber experiments for chemical reactions and does not include their fluctuations in solving the equations of motion for each time step. Further, the current operational regional-scale meteorological and air quality models do not explicitly simulate the stochastic nature of the atmosphere and, as such, typically miss the extreme values at both the low and high ends of the concentration distribution function.

In most applications of regional-scale air quality models, statistical metrics such as bias, root mean square error (RMSE), correlation, and index of agreement are being used to judge the quality of model predictions and determine if the model is suitable for forecasting or regulatory purposes (e.g., Fox, 1981, 1984; Solazzo et al., 2011; Appel et al., 2012; Simon et al., 2012; Foley et al., 2014; Ryan et al., 2016; Emery et al. 2016; Zhang, 2016; U.S. EPA, 2018). While significant improvements in the formulation, physical and chemical parameterizations, and numerical techniques have been implemented in atmospheric models over the past three-decades, it is not clear if the improvement claimed in the model's performance relative to the routine network measurements is statistically significant based on these metrics (Hogrefe et al., 2008). Also, no assessments have been made to date on the errors that are to be expected even from "perfect" regional-scale air quality modeling systems. To estimate such irreducible model errors due to atmospheric stochasticity (which we consider to be the errors that are expected even from a "perfect" model (devoid of structural and parametric uncertainties) with "perfect" (error-free) inputs), we analyzed the observed daily maximum 8-hr (DM8HR) ozone time series data at monitoring locations across the contiguous United States (CONUS) during the 1981-2014 time period and present the results of this analysis in Section 3.1. In Section 3.2, we illustrate how this information could be used in guiding model development specifically aimed at addressing reducible errors in the synoptic component by contrasting the results from Section 3.1 with analysis using the synoptic component from a 21-year simulation performed with the fully coupled WRF-CMAQ simulations covering the 1990-2010 period. Since we relied on multi-decadal historical ozone observations to assess the impact of the stochastic nature of the atmosphere, the results presented here are applicable to both forecasting and retrospective applications of current regional-scale air quality models.

## 2 Data and Methods

Ground-level DM8HR ozone data covering the CONUS during May to September in each year were obtained from the U.S. Environmental Protection Agency's (EPA) Air Quality System (AQS) (see https://www.epa.gov/aqs). A valid ozone season consists of at least 80% data coverage during May to September at each station. A total 185 monitoring stations with at least

30 valid years (to provide enough variety of synoptic conditions, denoted hereafter as 30+ in this paper) from the year 1981 to 2014 are analyzed. Also, fully coupled WRF-CMAQ model simulations over the CONUS for the 1990-2010 period were utilized in this study to demonstrate a new perspective on model performance evaluation. To ensure better characterization of the prevailing meteorology (i.e., synoptic forcing) in the retrospective 21-year WRF-CMAQ simulations, four-dimensional data assimilation (FDDA) was utilized following the methodology suggested by Gilliam et al. (2012) and modified for fully-coupled meteorology-chemistry model applications as described in Hogrefe et al. (2015). The model set-up and performance evaluation of these historical multiyear WRF-CMAQ simulations have been published by Xing et al. (2015), Gan et al. (2015), and Astitha et al. (2017). Time-varying chemical lateral boundary conditions are nested from the 108 km hemispheric WRF-CMAQ simulation from 1990 to 2010 (Xing et al., 2015).

It has been shown that time series of the daily maximum 8-hr ozone concentrations contain fluctuations operating on different time scales (e.g., intra-day forcing induced by the fast-changing emissions and atmospheric boundary layer evolution; diurnal forcing induced by the day and night differences; synoptic forcing induced by the passage of weather systems across the country, sub-seasonal forcing due to Madden-Julian Oscillation (MJO), and long-term forcing induced by emissions, El-Nino-Southern Oscillation (ENSO), climate change, and other slow-varying processes such as seasonal and sub-seasonal variations in the atmospheric deposition and stratosphere-troposphere exchange processes) as noted by Rao et al. (1997), Vukovich, (1997), Hogrefe et al. (2000), Porter et al. (2015), Astitha et al. (2017), Xing et al. (2016), and Mathur et al. (2017)). Variations in the 8-hour ozone can be thought of comprising of the baseline of pollution that is created by various emitting sources and modulated by the prevailing synoptic weather conditions (Rao et al., 1996 and 2011). Thus, the magnitude of the baseline (BL) concentration and the strength of the synoptic component (SY) should be viewed as the necessary and sufficient conditions for how high ozone levels can reach on a given day (Astitha et al., 2017). Scale separation can be achieved by applying filtering methods such as the Empirical Mode Decomposition (Huang et al., 1998), Elliptic filter (Poularika, 1998), Kolmogorov-Zurbenko (KZ) filter (Rao and Zurbenko, 1994), Adaptive Filter Technique (Zurbenko, et al., 1996), and Wavelet (Lau and Weng, 1995). Because Improved Complete Ensemble Empirical Mode Decomposition with Adaptive Noise (Improved CEEMDAN, a version of the Empirical Mode Decomposition method) and KZ filter yielded similar results for the DM8HR time series data as shown in Figs. 1-2 discussed in the next section, only the results from the KZ filter are presented in the subsequent analysis for quantifying the impact of the stochastic nature of the atmosphere on observed and simulated ozone concentrations. Furthermore, the KZ filtering is a simple method and works well even in the presence of missing data (Hogrefe et al., 2003). In this study, we used the $KZ_{5,5}$ with a window size of 5 days and 5 iterations on raw ozone time series [$O_3$ (t)] in the same manner as in Luo et al. (2019), Porter et al. (2015), and Rao et al. (2011). The size of the window and the number of iterations determine the desired scale separation. The $KZ_{5,5}$ filtering process helps separate the synoptic-scale weather-induced variations embedded in the May-September DM8HR time series data (short-term component, noted as SY) from the long-term baseline component (denoted as BL).

$$BL(t) = KZ_{5,5}\big(O_3(t)\big) \qquad (1)$$

$$SY(t) = O_3(t) - KZ_{5,5}(O_3(t)) \qquad (2)$$
$$O_3(t) = SY(t) + BL(t) \qquad (3)$$

Because we are working with the daily maximum 8-hr ozone data, the Nyquist interval is 2-days, indicating that the dynamical features having time scales less than 2 days (e.g., intra-day forcing from fast changing emissions and chemical transformations, boundary layer evolution, diurnal forcing due to night vs. day differences) are not resolvable in this analysis (see Fig. 2 in Dennis et al., 2010). The 50% cut-off frequency for the $KZ_{5,5}$ is ~24 days, and, hence, time scales less than those associated with synoptic-scale weather fluctuations are embedded in the short-term or SY forcing. The KZ filtering is applied to both DM8HR observations and modeled DM8HR time series. Once the baseline is separated from the original DM8HR time series from all monitoring stations, then the synoptic forcing in the historical ozone time series data is used to estimate the variability in ozone concentrations that can be expected because of the chaotic/stochastic nature of the atmosphere by taking into account the relationship between the strength of synoptic forcing and mean of baseline ozone at each location over CONUS. This methodology was applied to both measured and modeled ozone concentrations (see details in Luo et al., 2019). Whereas the focus of Luo et al. (2019) was on transforming the deterministic modeling results into a probabilistic framework for assessing the efficacy of different emission control strategies in achieving compliance with the ozone standard, this paper is aimed at quantifying the model performance errors to be expected at each monitoring site over CONUS even from "perfect" regional-scale ozone models driven with "perfect" input data from the ever-present stochastic nature of the atmosphere.

## 3 Results and Discussion

### 3.1 Analysis of ambient ozone data

Using both Improved CEEMDAN and KZ filtering methods, we separated the synoptic forcing (time scale < 24 days) and baseline (time scale > 1 month) forcing embedded in the time series of observed and modeled daily maximum 8-hour ozone concentrations. To illustrate, the results from the application of Improved CEEMDAN to the daily maximum 8-hr ozone time series data measured at Altoona, PA are presented in Fig. 1. The top left panel displays the raw ozone time series while the top of the right panel shows its power spectrum. The 7 intrinsic mode functions (IMFs) and the residual on the left side, and their corresponding power spectra on the right reveal that most of the synoptic-scale features in ozone data are imbedded in IMFs 1 and 2. The baseline ozone is extracted by removing the first two IMFs from the raw ozone time series. To illustrate the concept of the ozone baseline, DM8HR time series measured in 2010 at Altoona, PA is presented in Fig. 2a together with the embedded baseline concentration as extracted by the $KZ_{5,5}$ and Improved CEEMDAN. It is evident that high ozone levels are always associated with the elevated baseline. The difference between the raw ozone time series and baseline, denoted as the short-term or synoptic forcing (SY), is displayed in Fig. 2b. The power spectra, displayed in Figs. 2c and d, reveal both methods yielded good scale separation. Due to the good agreement between both scale separation techniques, only the results from the KZ filter are presented for the remainder of the manuscript.

Once the scale separation is achieved with the $KZ_{5,5}$, we superimposed the SY forcing imbedded in 30+ years of historical DM8HR ozone time series measured at a given location on the baseline component of the ozone time series at that location to generate 30+ reconstructed or pseudo ozone distributions. Illustrative results using eq. (3) at a suburban location in Altoona, PA are presented for 2010 base year in Fig. 3a; it should be noted that the linear relationship between the strength of SY (defined as the standard deviation of the data in the synoptic component) and the magnitude of the BL (defined as the mean of the data in the baseline component) has been taken into account in generating 30+ years of adjusted SY forcing as illustrated in Luo et al. (2019). As expected, there is excellent agreement between the average of 30+ values (solid blue line) and observed ozone in 2010 at each percentile of the concentration distribution function (red line). Also, the original cumulative distribution function (CDF) in 2010 (red line) is constrained within the 30+ CDFs of pseudo distributions (Fig. 3a); note, it is equally likely for any of these 30+ CDFs to occur because of the stochastic nature of the atmosphere even though the individual event in 2010 yielded the CDF shown in red. As mentioned before, ozone mixing ratio at any given probability point on the red line in Fig. 3a reflects an individual event while ozone values at the same probability in different CDFs (gray lines) reflect the population stemming from the stochastic nature of the atmosphere. In other words, there are 30+ dynamically consistent ozone time series attributable to the 2010 baseline (given 2010 emissions) for examining the inherent variability due to atmospheric stochasticity. It is evident in Fig. 3a that there is larger variability at the lower and upper percentiles than that in inter-quartile range, revealing that the tails of the concentration distribution function are subject to large inherent uncertainty. Using these 30+ pseudo-observation ozone mixing ratios and the actual observed ozone values at each percentile, statistical metrics such as Bias, RMSE, coefficient of variation (CV=standard deviation/mean), normalized mean error (NME) and normalized mean bias (NMB) are presented in Fig. 3b and c (see Emery et al. (2016) for the description of the statistical metrics considered here). As expected, the lower and upper tails of the distribution are prone to large errors. These results demonstrate the presence of substantial natural variability at the upper 95th percentile, which is of primary interest in regulatory analyses. The extreme values are better described in statistical terms rather than in deterministic sense (Hogrefe and Rao, 2001; Luo et al., 2019).

Ozone time series at 185 monitoring stations covering CONUS, having at least 80% data completeness, are analyzed in the above manner and the results are displayed as box plots in Fig. 4. Note the presence of large variability in the CV, NME, and NMB, and Bias at the lower and upper percentiles (Fig. 4). The RMSE expected for the ozone mixing ratios in the interquartile range is ~1.5 ppb, but it is >5 ppb for the upper 95th percentile (Fig. 4b). The spatial distribution of RMSE at the 50th and 95th percentiles is displayed in Figures 5a and 5b, respectively. The RMSE at the upper 95th percentile is very high at some monitoring sites in California and Michigan (Fig. 5b). Monitoring stations situated in the urban areas, near large water bodies, and in regions of complex terrain influenced predominantly by local conditions tend to exhibit higher RMSE. The elevation of the monitoring sites is displayed in Fig. 5c.

### 3.2 Analysis of modeled ozone concentrations

The analysis in the previous section quantified the inherent stochastic variability that is present in the SY component using long-term records of ozone observations. In this section, we analyze long-term records of model simulations in an attempt to quantify the error associated with the modeled SY component that results both from not explicitly representing stochastic variations in atmospheric dynamics in the current generation regional air quality models and from other reducible sources of model error. The model simulations were performed with the fully coupled WRF-CMAQ system with a 36-km horizontal grid cell size and covered the 21-year period from 1990 to 2010 (Gan et al., 2015). In this section, we examine the impact of superimposing different SY forcings embedded in ozone observations vs. those in the WRF-CMAQ model on the observed baseline concentration. To provide an illustration of the differences between observed and modeled time series over this period, Figure 6a displays a scatter plot of the strength of the SY component (standard deviation of data in the SY component) vs. the mean of the baseline (BL) component for both observations and model simulations at the Altoona, PA site. While both observations and WRF-CMAQ simulations show a strong correlation between these two variables, it is evident that at this monitoring location the standard deviation (i.e., strength) of the SY component is substantially lower for the WRF-CMAQ simulations for a given mean of the BL component (i.e., for any given year). The year-to-year variation in the observed and modeled mean of BL and strength of SY forcing, displayed in Fig. 6b, reveals that the model overestimated BL and underestimated the strength of SY forcing. The 36-km grid may be better representing the large-scale synoptic forcing associated with the translation of weather systems than the meso-scale weather and urban influences (both dynamics and chemistry) that are embedded in the observed SY component. Meteorological modeling with higher horizontal grid resolution might be able to capture the land-sea breeze, lake-sea breeze, and terrain influences that observations are seeing at certain monitoring locations.

To isolate the impact of model imperfections on only the SY time scale on errors across the ozone distribution, we assume that the model perfectly reproduces the 'true' BL depicted by the observed 2010 BL. We then use this 'perfect' modeled BL and reconstruct 'pseudo-simulated' ozone time series, like what was done in Fig. 3, except for using the SY component embedded in the 21 years of coupled WRF-CMAQ simulations. The rationale for this analysis is to quantify the amount of model error present in the current simulations that could conceivably be reduced through improving the representation of synoptic and mesoscale processes and/or increased horizontal resolution with appropriate data assimilation techniques. Fig. 7a displays the CDF of actual observed ozone (red line) overlaid on 21 pseudo-simulated ozone CDFs (gray lines, with averages of all 21 pseudo-simulated ozone percentiles shown in blue) at the Altoona, PA site while Figs. 7b and 7c display absolute and normalized performance metrics. Figure 7a confirms that the coupled WRF-CMAQ SY components have less intra-annual variability than observed SY components, causing overestimation at the low end and underestimation at the high end of the observed CDF for all 21 years of reconstruction; these results imply that the model's results at the upper and lower percentiles will always tend to be unreliable or prone to large errors even when the baseline concentration is predicted perfectly. The U-

shape of the absolute and relative error curves in Figures 7b and c is similar to the corresponding curves in Figure 3, but the larger magnitude at the high and low end of the distribution indicates that the effects of the underestimated intra-annual SY variability (note that the distribution of modeled values in Fig. 7a is much flatter (i.e., having higher Kurtosis) than that of the observations) outweigh those errors attributable to the stochastic variability presented in Figure 3. The shape of the absolute and normalized bias curves deviates from those shown for the pseudo-observations in Figures 3b-c and, thus, also reveals the effect of the underestimation of the intra-annual SY variability. Figures 7d-f present differences between the curves shown in Figures 7a-c and a version of Figure 3a-c computed from the 1990-2010 data instead of 30+ years of historical ozone observations. Panels e and f show that at the 50th percentile, the differences in the error curves are close to zero since both the pseudo-simulations and pseudo-observations used the same observed BL component. At the upper percentiles, the differences reach 3 – 5 ppb, providing an estimate of the reducible error in simulating the extreme values at this location because of the differences in the observed SY and WRF-CMAQ SY components at this location; high-resolution meteorological modeling may help address these reducible errors.

Figs. 8a and b display the RMSE at the median and 95th percentile for the 'pseudo-simulated' ozone values at each monitoring site. For the 50th percentile, the RMSE values range from 0.2 ppb to 3.2 ppb over CONUS with a median value of 1 ppb while at the 95th percentile, the RMSE values range from 1 ppb to 15 ppb with a median value of 4 ppb across all sites over CONUS. The values are highest along the California coast and near Great Lakes, possibly due to inadequacies in simulating the land-sea breeze and land-lake breeze regimes, respectively, with modeling at 36 km grid cells. Air quality modeling uncertainty even for the retrospective modeling cases, outside of the chemistry formulation and boundary conditions, is attributed primarily to meteorology and emissions inputs. Vautard et al. (2012) and McNider and Pour-Biazar (2020) concluded that major challenges remain in the simulation of prevailing meteorology (e.g., errors in wind speed, PBL, night-time meteorology, nocturnal transport aloft, clouds) in retrospective air quality modeling. Based on the retrospective ozone episodic modeling with the WRF-CMAQ model using various sets of equally likely initial conditions for meteorology along with FDDA, Gilliam et al. (2015) confirmed the presence of sizable spread in WRF solutions, including common weather variables of temperature, wind, boundary layer depth, clouds, and radiation, thereby causing a relatively large range of ozone concentrations. Also, pollutant transport is altered by hundreds of kilometers over several days. Ozone concentrations of the ensemble varied as much as 10–20 ppb (or 20–30%) in areas that typically have higher pollution levels. As model improvements are made, one can quantitatively assess how close the predictions of the improved model are for each percentile for the given base year simulation to the expected errors from a "perfect" model with "perfect" input, i.e. the target RMSE shown in Fig.5a and b. Perhaps, the next generation of regional-scale meteorological and air quality models might be capable of explicitly simultaneously treating the mean and fluctuation components for all variables within the deterministic-stochastic modeling framework to properly account for the stochastic nature of the atmosphere.

**4 Conclusions**

Regardless of how accurate the regional air quality model is, the stochastic variations in the atmosphere cannot be consistently reproduced by the deterministic numerical models. In this study, we demonstrate how to quantify this irreproducible stochastic component by isolating the synoptic forcing imbedded in 30+ years of historical observations and assess the performance of the 36 km fully coupled WRF-CMAQ model in simulating 21 years of ozone concentrations over the contiguous U.S. Observation-based analysis reveals that on average, the irreducible error attributable to the stochastic nature of the atmosphere ranges from ~2 ppb at the $50^{th}$ percentile to ~ 5 ppb at the $95^{th}$ percentile. To improve regional-scale ozone air quality models, attention should be paid to accurately simulate the baseline concentration by focusing on the quality of the emission inventory and the model's treatment for the boundary conditions and slow-changing (operating on sub-seasonal, seasonal, and longer-term time scales) atmospheric processes. Also, errors in reproducing the synoptic forcing can possibly be reduced with high-resolution meteorological modeling using appropriate data assimilation techniques. Nonetheless, these results demonstrate the presence of large variability in the upper tail of the DM8HR $O_3$ concentration cumulative distribution even with "perfect" models using "perfect" input data. Having this quantitative estimation of practical limits for model's accuracy helps in objectively assessing the current state of regional-scale air quality models, measuring progress in their evolution, and providing meaningful and firm targets for improvements in their accuracy relative to measurements from routine networks.

**Code availability**: Source code for version 5.0.2 of the Community Multiscale Air Quality (CMAQ) modeling system can be downloaded from https://github.com/USEPA/CMAQ/tree/5.0.2. For further information, please visit the U.S. Environmental Protection Agency website for the CMAQ system: https://www.epa.gov/cmaq.

**Data availability**: All ozone observations used in this article are available from https://aqs.epa.gov/aqsweb/airdata/download_files.html (AQS). Paired ozone observation and CMAQ model data used in the analysis will be made available at https://edg.epa.gov/metadata/catalog/main/home.page. Raw CMAQ model outputs are available on request from the U.S EPA authors.

**Competing interests**: The authors declare that they have no conflict of interest.

**Author Contribution:** STR conceptualized the idea. STR, CH, VG, and RM designed the analysis approach. CH and RM post-processed previously conducted model simulations. HL performed data analyses and prepared the illustrations. STR prepared the manuscript with contributions from all co-authors.

**Disclaimer:** The views expressed in this paper are those of the authors and do not necessarily represent the view or policies of the U.S. Environmental Protection Agency.

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

**List of Figures**

Figure 1. Results of the application of the Improved CEEMDAN technique (a modified version of EMD) technique, which is designed for analyzing non-stationary and non-linear time series data, to the daily maximum 8-hour ozone time series data at the Altoona, PA site. The numbers on the right side represent the time scale (in days) associated with each IMF. Note, the power spectrum of raw ozone time series (upper right panel) shows that the energy in the 1-10 days (synoptic) time scale is an order of magnitude less than that in the longer (baseline) time scale.

Figure 2a. Raw observed DM8HR ozone time series (black) and the embedded baseline (red for EMD and blue for KZ) at Altoona, PA in 2010; Figure 2b. Time series of synoptic forcing (red for EMD and blue for KZ); Figure 2c and 2d are their corresponding power spectra. The bottom two panels compare the power spectra of the baseline forcing (left) and the synoptic forcing (right) derived from KZ filtering and EMD (sum of IMF 1 and IMF2). Notice that most of the energy in the baseline time series is in the longer time scale while most of the energy of the short-term component is in the high-frequency range. The similarity of results from both scale separation techniques demonstrates that the two scales of interest (i.e., baseline and synoptic forcing) have been extracted reasonably well by these two methods.

Figure 3a: Comparison between the observed cumulative distribution function (CDF) for 2010 shown in red with 30+ pseudo-observations CDFs generated from historical DM8HR ozone time series shown in gray at a suburban site at Altoona in PA (AQS station identifier 420130801). The blue line represents the average of the 30+ gray lines; Figure 3b: Display of various statistical metrics (standard deviation (std), root mean square error (RMSE), bias) derived by comparing the actual observed and pseudo ozone values in Fig. 3a; Figure 3c: Normalized statistical metrics of normalized mean error (NME), normalized mean bias (NMB), coefficient of variation (CV). Notice the large variability occurring at the lower and upper percentiles.

Figure 4. Box plots of statistical metrics based on the results from the analysis of DM8HR data at 185 monitoring sites: (a) Standard deviation, (b) Root mean square error, (c) Mean bias, (d) Coefficient of variation, (e) Normalized mean error, and (f) Normalized mean bias. The lower and upper edges of the boxes represent the 25th and 75th percentile values while the whiskers represent the 5th and 95th percentiles. See data analysis procedures using the ozone baseline observed in the year 2010 as the target BL in equations 7 and 8 of Luo et al. (2019).

Figure 5. Spatial distribution of the lower bound for the RMSE or expected RMSE at each monitoring site over CONUS (a) at the median and (b) at the 95th percentile; (c) elevation (km) above the mean sea level of each monitoring site.

Figure 6. (a) Scatter plot of the standard deviation (i.e., strength) of the SY component vs. the mean of the baseline (BL) component for each of the 21 years from 1990 to 2010 at the Altoona, PA monitoring site. Observations are shown in red while

WRF-CMAQ results are shown in blue. (b) Inter-annual variability in the mean of the baseline component and standard deviation of the synoptic component in the WRF-CMAQ model and observations at the Altoona, PA site. Although year-to-year variation is captured, the model has overestimated the baseline forcing and underestimated the synoptic forcing.

5    Figure 7.  a) Comparison between the observed CDF overlaid on 21 'pseudo-simulated' or reconstructed ozone CDFs with SY generated from modeled DM8HR ozone time series at a suburban site at Altoona in PA (AQS station identifier 420130801); b) Display of various statistical metrics derived by comparing the actual observed and pseudo-simulated ozone values in Fig. 7a; c) Normalized statistical metrics; d).Difference between the pseudo-simulated CDFs shown in Figure 7a and the pseudo-observed CDFs as shown in Figure 7a but calculated from 21 years (1990-2010) of observations only. The gray lines represent

10    the differences for a specific SY year while the blue line represents the differences between the means of the 21 reconstructions; e) Difference between the absolute performance metrics for pseudo-simulations shown in Figure 7b and those calculated for pseudo-observations as shown in Figure 7b but calculated for 21 years (1990-2010) only. f) As in panel e) but for normalized performance metrics.

15    Figure 8. Errors in the 21 'pseudo-simulated' or reconstructed ozone time series with SY generated from modeled DM8HR ozone time series using BL obtained from observations at (a) the median and (b) 95th percentile.

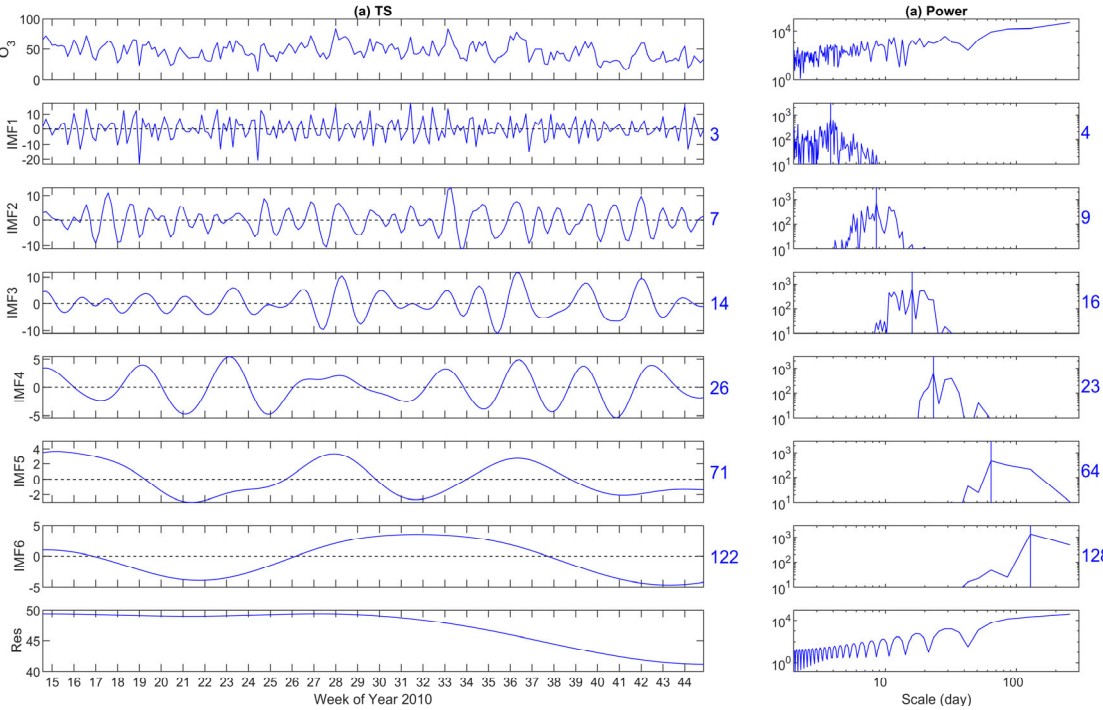

**Figure 1. Results of the application of the Improved CEEMDAN technique (a modified version of EMD) technique, which is designed for analyzing non-stationary and non-linear time series data, to the daily maximum 8-hour ozone time series data at the Altoona, PA site. The numbers on the right side represent the time scale (in days) associated with each IMF. Note, the power spectrum of raw ozone time series (upper right panel) shows that the energy in the 1-10 days (synoptic) time scale is an order of magnitude less than that in the longer (baseline) time scale.**

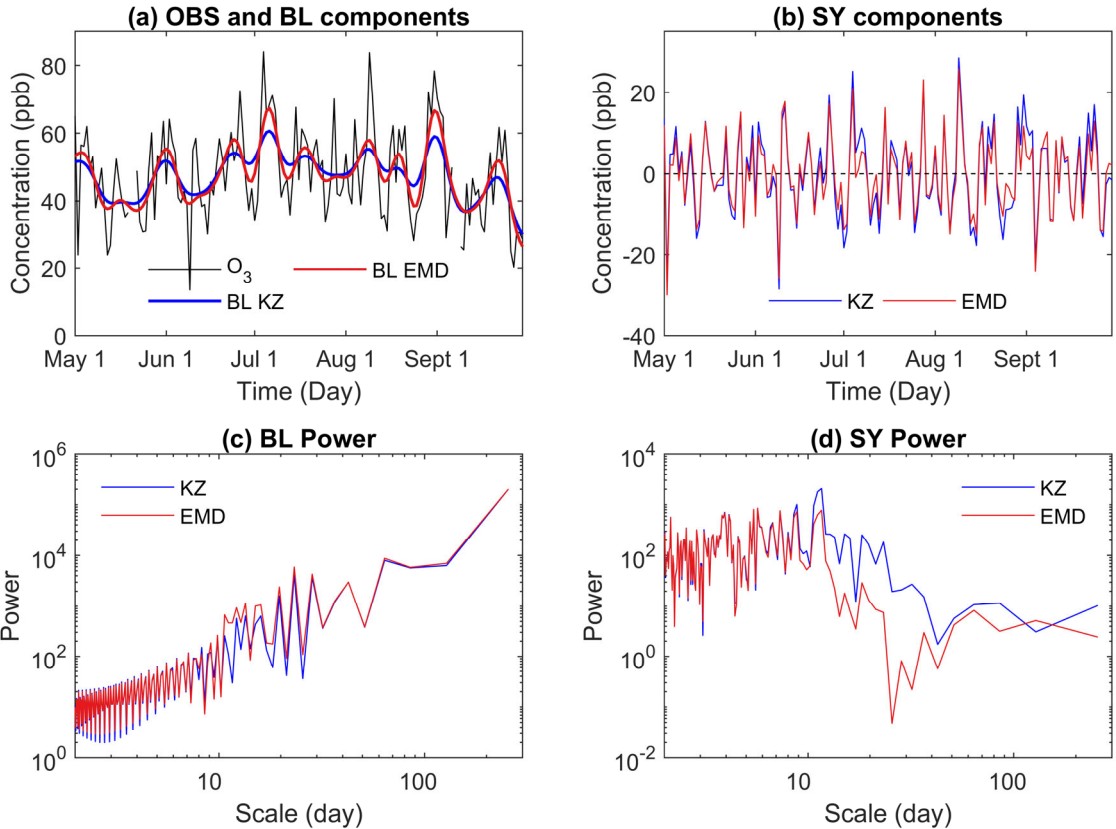

**Figure 2a.** Raw observed DM8HR ozone time series (black) and the embedded baseline (red for EMD and blue for KZ) at Altoona, PA in 2010; **Figure 2b.** Time series of synoptic forcing (red for EMD and blue for KZ); **Figure 2c and 2d** are their corresponding power spectra. The bottom two panels compare the power spectra of the baseline forcing (left) and the synoptic forcing (right) derived from KZ filtering and EMD (sum of IMF 1 and IMF2). Notice that most of the energy in the baseline time series is in the longer time scale while most of the energy of the short-term component is in the high-frequency range. The similarity of results from both scale separation techniques demonstrates that the two scales of interest (i.e., baseline and synoptic forcing) have been extracted reasonably well by these two methods.

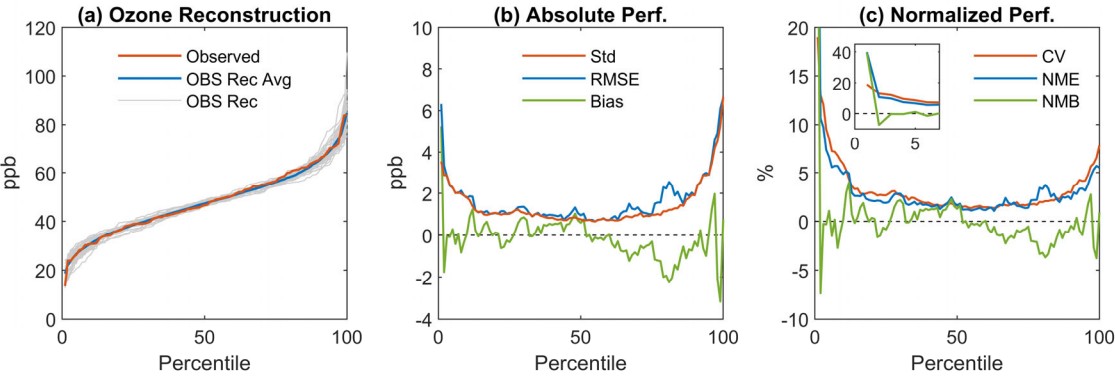

**Figure 3a: Comparison between the observed cumulative distribution function (CDF) for 2010 shown in red with 30+ pseudo-observations CDFs generated from historical DM8HR ozone time series shown in gray at a suburban site at Altoona in PA (AQS station identifier 420130801). The blue line represents the average of the 30+ gray lines; Figure 3b: Display of various statistical metrics (standard deviation (std), root mean square error (RMSE), bias) derived by comparing the actual observed and pseudo ozone values in Fig. 3a; Figure 3c: Normalized statistical metrics of normalized mean error (NME), normalized mean bias (NMB), coefficient of variation (CV). Notice the large variability occurring at the lower and upper percentiles.**

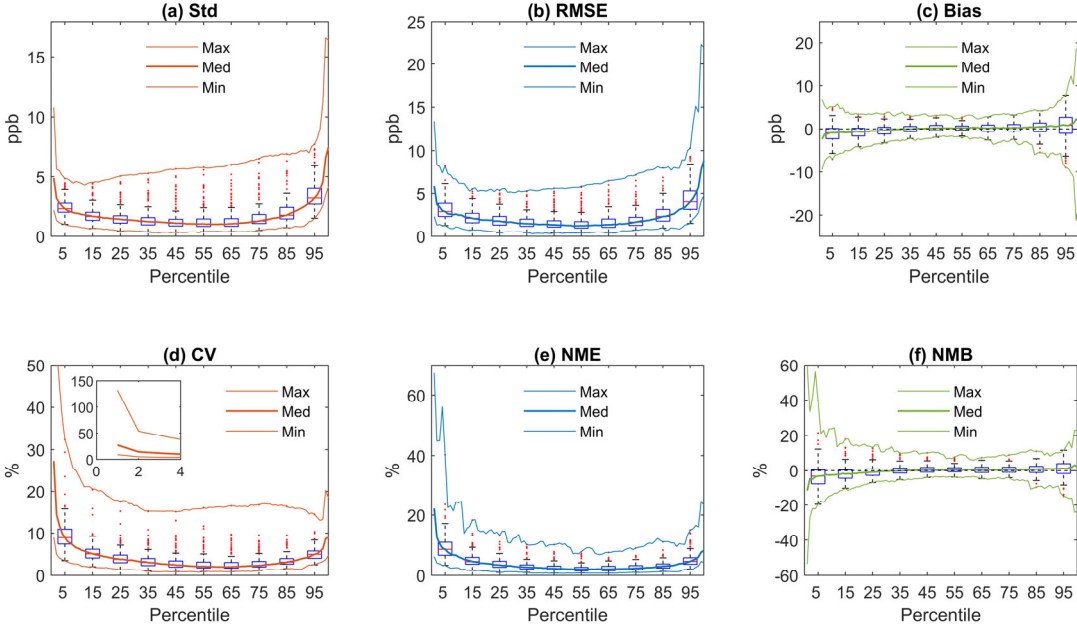

**Figure 4. Box plots of statistical metrics based on the results from the analysis of DM8HR data at 185 monitoring sites: (a) Standard deviation, (b) Root mean square error, (c) Mean bias, (d) Coefficient of variation, (e) Normalized mean error, and (f) Normalized mean bias. The lower and upper edges of the boxes represent the 25th and 75th percentile values while the whiskers represent the 5th and 95th percentiles. See data analysis procedures using the ozone baseline observed in the year 2010 as the target BL in equations 7 and 8 of Luo et al. (2019).**

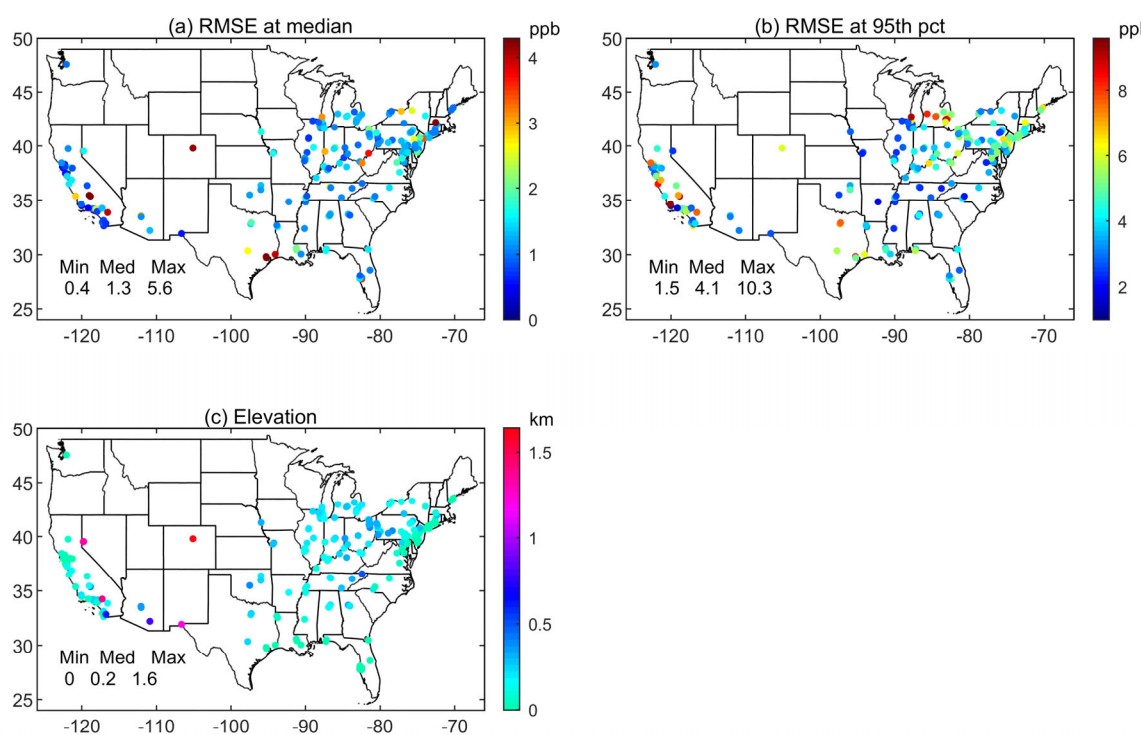

5   **Figure 5. Spatial distribution of the lower bound for the RMSE or expected RMSE at each monitoring site over CONUS (a) at the median and (b) at the 95th percentile; (c) elevation (km) above the mean sea level of each monitoring site.**

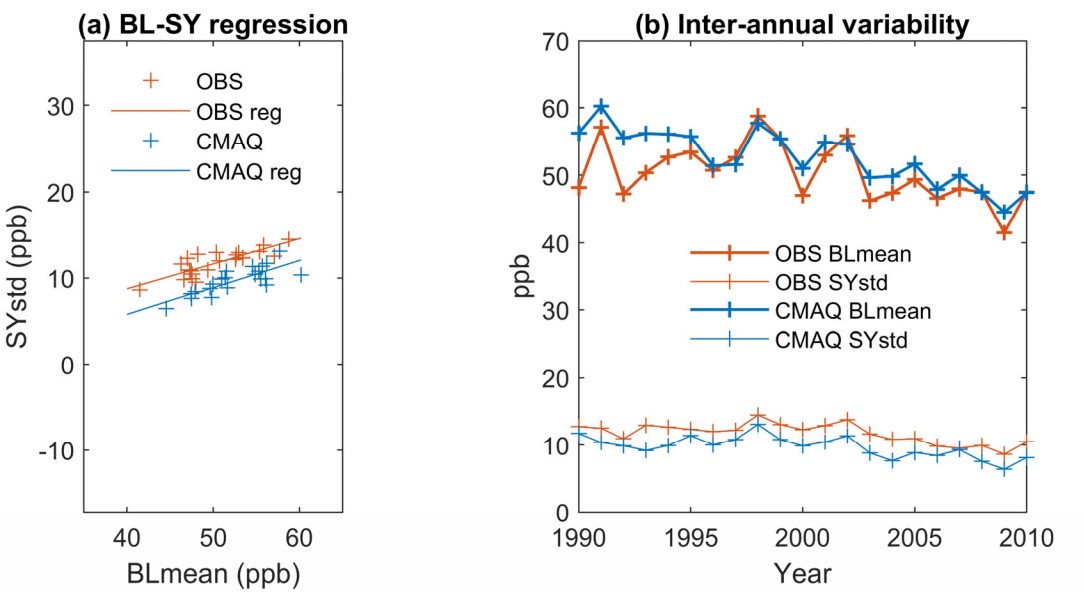

**Figure 6. (a) Scatter plot of the standard deviation (i.e., strength) of the SY component vs. the mean of the baseline (BL) component for each of the 21 years from 1990 to 2010 at the Altoona, PA monitoring site. Observations are shown in red while WRF-CMAQ results are shown in blue. (b) Inter-annual variability in the mean of the baseline component and standard deviation of the synoptic component in the WRF-CMAQ model and observations at the Altoona, PA site. Although year-to-year variation is captured, the model has overestimated the baseline forcing and underestimated the synoptic forcing.**

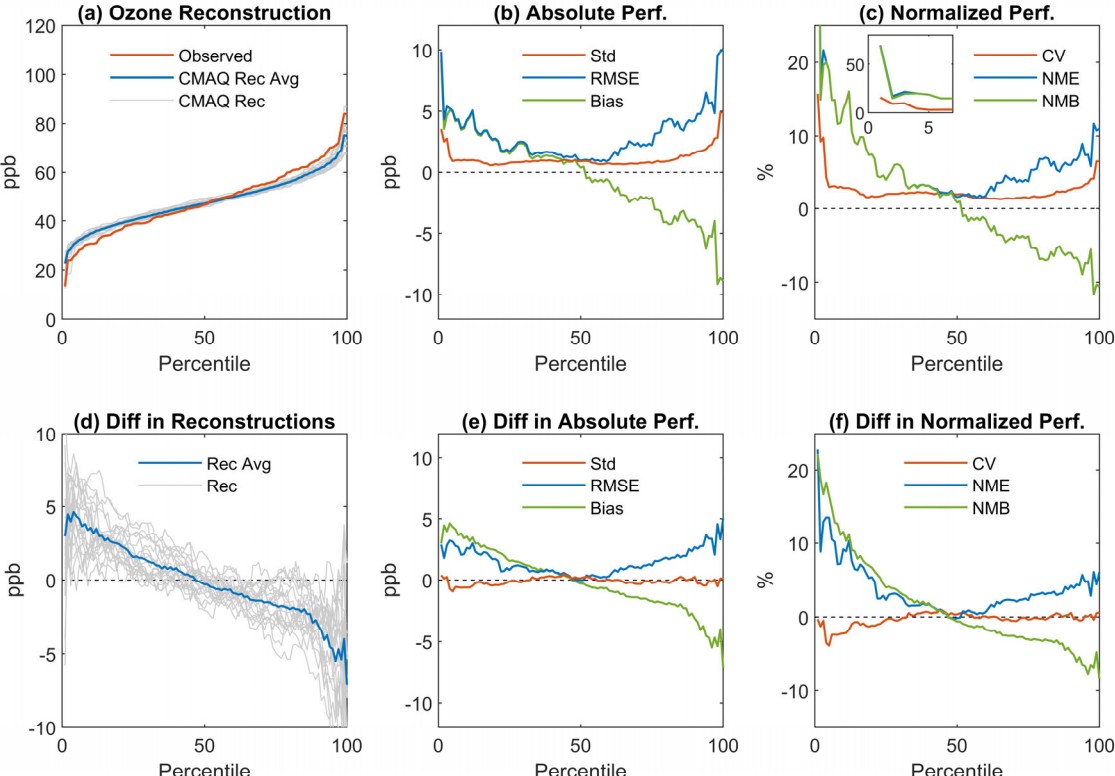

**Figure 7.** **a) Comparison between the observed CDF overlaid on 21 'pseudo-simulated' or reconstructed ozone CDFs with SY generated from modeled DM8HR ozone time series at a suburban site at Altoona in PA (AQS station identifier 420130801); b) Display of various statistical metrics derived by comparing the actual observed and pseudo-simulated ozone values in Fig. 7a; c) Normalized statistical metrics; d).Difference between the pseudo-simulated CDFs shown in Figure 7a and the pseudo-observed CDFs as shown in Figure 7a but calculated from 21 years (1990-2010) of observations only. The gray lines represent the differences for a specific SY year while the blue line represents the differences between the means of the 21 reconstructions; e) Difference between the absolute performance metrics for pseudo-simulations shown in Figure 7b and those calculated for pseudo-observations as shown in Figure 7b but calculated for 21 years (1990-2010) only. f) As in panel e) but for normalized performance metrics.**

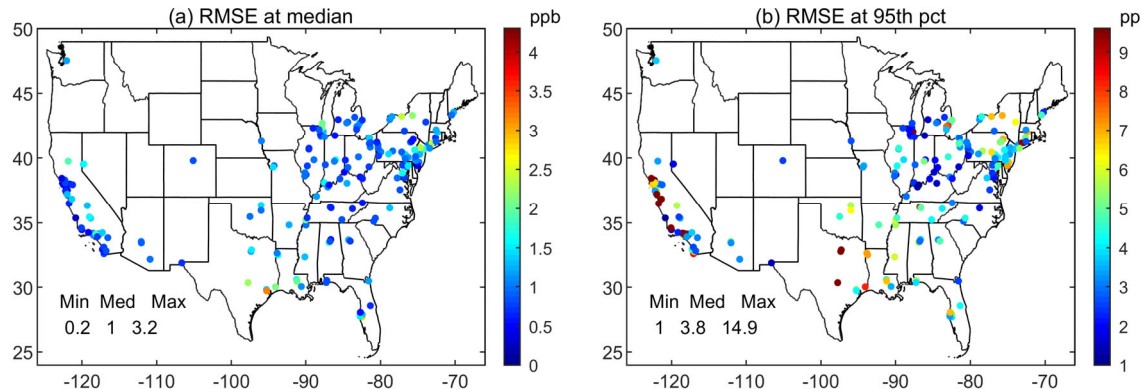

**Figure 8. Errors in the 21 'pseudo-simulated' or reconstructed ozone time series with SY generated from modeled DM8HR ozone time series using BL obtained from observations at (a) the median and (b) 95th percentile.**