# Peer review of "On the Limit to the Accuracy of Regional-Scale Air Quality Models"

_Atmospheric Chemistry and Physics, 2019_

## Referee Comment (RC1) · Anonymous Referee #1 · 19 Sep 2019

This manuscript separated short-term synoptic-scale fluctuations from long-term baseline component embedded in the daily maximum 8-hr ozone time series using a filter and estimated the limit of air quality model's accuracy (or predictability/uncertainties of air quality prediction). This is an interesting topic for air quality prediction.

But to my surprise, the authors did not even consider lead time when discussing air quality predictability (or limit of air quality prediction). What is the configuration of the air quality prediction? Was this one-day prediction? Two-day prediction? Prediction uncertainties/errors will change significantly with different lead time.

It is also surprising to see the authors suggesting improving simulation of the baseline concentration by focusing on the quality of the emission inventory and the model's treatment for the slow-changing atmospheric processes. I have no question for improv-

ing emission inventory, but I am confused by improving the slow-changing atmospheric processes. The huge advances of weather prediction during the past few decades has been focusing on 1-day or 2-day prediction. On such short-term synoptic-scale weather processes, our weather prediction did excellent job and has been improved through years. Such improvement can benefit air quality prediction significantly (Zhang et al., 2007).

I would thus imagine that such short-term practical predictability of air quality can be much improved through better model treatments and better initial conditions of meteorological and chemical variables, as well as emissions.

Other specific comments:

Many of the concept/discussion regarding inherent/practical predictability, reducible/irreducible uncertainties are questionable/wrong, or different from those used in weather prediction. For example, emissions are definitely reducible uncertainties and factors in practical predictability. Please carefully define those terms/concepts and refer to normally used/accepted definitions.

The writing is overly concise, particularly in many cases where detailed explanation is needed.

References:

Zhang, F., Bei, N., Nielsen-Gammon, J. W., Li, G., Zhang, R., Stuart, A., & Aksoy, A. (2007). Impacts of meteorological uncertainties on ozone pollution predictability estimated through meteorological and photochemical ensemble forecasts. Journal of Geophysical Research, 112, D04304. https://doi.org/10.1029/2006JD007429

---

## Author Comment (AC1) · 3 Oct 2019

**Authors' Responses (in red) to the Comments on acp-2019-642 by Anonymous Referee # 1**

This manuscript separated short-term synoptic-scale fluctuations from long-term baseline component embedded in the daily maximum 8-hr ozone time series using a filter and estimated the limit of air quality model's accuracy (or predictability/uncertainties of air quality prediction). This is an interesting topic for air quality prediction.

The authors thank the referee for recognizing the importance of our work.

But to my surprise, the authors did not even consider lead time when discussing air quality predictability (or limit of air quality prediction). What is the configuration of the air quality prediction? Was this one-day prediction? Two-day prediction? Prediction uncertainties/errors will change significantly with different lead time.

It seems that the referee has misunderstood the modeling simulations we have examined in this paper. Please note that our paper focused on the evaluation of retrospective simulations of 21-years (1990 to 2000) of the daily maximum 8-hour average ozone concentrations over the contiguous United States (CONUS) with the fully-coupled Weather Research Forecasting (WRF) meteorological model and the Community Multiscale Air Quality (CMAQ) chemical transport model. While the concepts presented in our paper are applicable to examining air quality forecasting products, the results of our study reflect the prediction capability of the model based on retrospective simulations and **not** air quality forecasting. To ensure better characterization of the prevailing meteorology (i.e., synoptic forcing) for these retrospective 21-year simulations, four-dimensional data assimilation (FDDA) was utilized following the methodology suggested by Gilliam et al. (see *Atmos. Environ*., Vol.53, pp 186-201, 2012) and modified for fully-coupled meteorology-chemistry model applications as described in Hogrefe et al. (see *Atmos. Environ.*, Vol. 115, pp 683-694, 2015). The modeling set-up and performance evaluation of these historical multiyear WRF-CMAQ simulations have been published by Xing et al. (2015), Gan et al. (2015), and Astitha et al. (2017) as referenced in our manuscript. The usage of FDDA will be clarified in finalizing the manuscript.

It is also surprising to see the authors suggesting improving simulation of the baseline concentration by focusing on the quality of the emission inventory and the model's treatment for the slow-changing atmospheric processes. I have no question for improving emission inventory, but I am confused by improving the slow-changing atmospheric processes.

A number of papers have been published, documenting the importance of the baseline (longer-term) forcing embedded in ambient ozone data (see e.g., Rao et al., 1996; Hogrefe et al., 2000; Rao et al., 2011; Porter et al., 2017; Astitha et al., 2017; Luo et al., 2019). As stated in our paper, the baseline level and the strength of the synoptic forcing are to be viewed as the necessary and sufficient conditions for observing peak ozone levels. Rao et al. (2011), Hogrefe et al. (2000), Astitha et al. (2017), Porter et al (2017), and Luo et al. (2019) have demonstrated that when the magnitude of the baseline level is low, there will not be ozone exceedances of the USA's National Ambient Air Quality Standard no matter how strong the synoptic forcing is. In this study, we are working with the model (WRF-CMAQ)-predicted and observed time series of the daily maximum 8-hour ozone concentrations during 1990 to 2000 at various

monitoring locations over CONUS; therefore, the Nyquist interval here is 2-days.  Using both Empirical Mode Decomposition (EMD) and KZ filtering, we separated the synoptic forcing (time scale < 24 days) and baseline (time scale > 1 month) forcing embedded in the time series of observed and modeled daily maximum 8-hour ozone concentrations.  To illustrate, the results of the application of EMD to the daily maximum 8-hr ozone time series data measured at Altoona, PA are presented in Fig. 1 below.  The top left panel displays the raw ozone time series while the top of the right panel shows its power spectrum.  The 7 intrinsic mode functions (IMFs) and the residual on the left side, and their corresponding power spectra on the right reveal that most of the synoptic-scale features in ozone data are reflected in IMFs 1 and 2.  The baseline ozone is extracted by removing the first two IMFs from the raw ozone time series.

[Figure]

Figure 1.  Results of the application of the EMD technique, which is designed for analyzing non-stationary and non-linear time series data, to the daily maximum 8-hour ozone time series data at the Altoona, PA site.  The numbers on the right side represent the time scale (in days) associated with each IMF.  Note,the power spectrum of raw ozone time series shows that the energy in the 1-10 days (SY forcing) is an order of magnitude less than that in the longer (baseline) time scale.

The scale separation achieved from the application of EMD and KZ filter, displayed in Fig. 2, reveal that the results are quite similar.  It should be noted that the short-term component (SY) extracted from the KZ filter as well as EMD's high-frequency IMFs 1 and 2 resemble white noise process.  This material will be included in finalizing the manuscript.

[Figure]

Figure 2.  The top left panel displays the raw daily maximum 8-hour ozone time series together with the baselines extracted from the KZ filter and EMD while the top right panel reveals the similarity between white noise and synoptic forcing imbedded in these observed ozone time series.  The bottom two panels compare the power spectra of the baseline forcing (left) and the synoptic forcing (right) derived from KZ filtering and EMD (sum of IMF 1 and IMF2).  Notice that most of the energy in the baseline time series is in the longer time scale while most of the energy of the short-term component is in the high-frequency range.  The similarity of results from both scale separation techniques demonstrates that the two scales of interest (i.e., baseline and synoptic forcing) have been extracted reasonably well.

The huge advances of weather prediction during the past few decades has been focusing on 1-day or 2-day prediction. On such short-term synoptic-scale weather processes, our weather prediction did excellent job and has been improved through years. Such improvement can benefit air quality prediction significantly (Zhang et al., 2007). I would thus imagine that such short-term practical predictability of air quality can be much improved through better model treatments and better initial conditions of meteorological and chemical variables, as well as emissions.

As stated before, our study deals with retrospective air quality simulations, not with results from an air quality forecasting effort.  Air quality modeling uncertainty even for the retrospective modeling cases, outside of the chemistry formulation, is attributed primarily to meteorology and emissions inputs. Vautard et al. (*Atm. Environ*., Vol. 53, pp 15-37, 2012) concluded that major challenges still remain in the simulation of prevailing meteorology (e.g., errors in wind speed, PBL, nighttime meteorology, clouds) in

retrospective air quality modeling. Based on retrospective ozone episodic modeling with the WRF-CMAQ model using various sets of equally likely initial conditions for meteorology along with FDDA, Gilliam et al. (2015) confirmed the presence of sizable spread in WRF solutions, including common weather variables of temperature, wind, boundary layer depth, clouds, and radiation, thereby causing a relatively large range of ozone concentrations. Also, pollutant transport is altered by hundreds of kilometers over several days. Ozone concentrations of the ensemble varied as much as 10–20 ppb (or 20–30%) in areas that typically have higher pollution levels.

We acknowledge that air quality forecasting often is concerned with accurately predicting transient pollution events. The analysis of modeling uncertainty in our retrospective simulations that employed data assimilation suggest that the largest improvement in forecast accuracy can be achieved through implementing bias correction techniques (i.e. by correcting for systematic errors such as those caused by emission uncertainties that manifest themselves primarily in the baseline) as demonstrated by Kang et al. (2008 *JGR-Atmospheres*, Vol. 113, Issue D23308; 2010 *Atmos. Environ.*, Vol. 44, 18, pp 2203-2212). However, we do not dispute the fact that any advances in predicting short-term atmospheric processes or phenomena, be it through model improvements, better initial conditions, or ensemble techniques, may help to reduce the portion of the prediction error that is not due to systematic biases. Rather, we argue through our analysis of both observations and retrospective air quality simulations that, at least for retrospective air quality planning applications, the focus of model development and evaluation efforts should be on longer time scales. This discussion will be added in finalizing the manuscript.

Other specific comments: Many of the concept/discussion regarding inherent/practical predictability, reducible/irreducible uncertainties are questionable/wrong, or different from those used in weather prediction. For example, emissions are definitely reducible uncertainties and factors in practical predictability. Please carefully define those terms/concepts and refer to normally used/accepted definitions. The writing is overly concise, particularly in many cases where detailed explanation is needed. References: Zhang, F., Bei, N., Nielsen-Gammon, J. W., Li, G., Zhang, R., Stuart, A., & Aksoy, A. (2007). Impacts of meteorological uncertainties on ozone pollution predictability estimated through meteorological and photochemical ensemble forecasts. Journal of Geophysical Research, 112, D04304. https://doi.org/10.1029/2006JD007429 Interactive comment on Atmos. Chem. Phys. Discuss., https://doi.org/10.5194/acp-2019-642, 2019.

It is difficult to identify exactly which terms the reviewer is referring to in comment that terms are not defined or are not defined as "normally used/accepted". From our perspective, we are consistent with the reviewer's example (we agree that uncertainty in retrospectively simulated ozone concentrations can be improved with improvement in emissions characterizations). The confusion in the definition of terms may be more to do with the reviewer's misunderstanding that the manuscript was focused on air quality forecasting rather than on analysis of retrospective simulations. We agree that terminologies between the weather forecasting community and the air quality modeling community may differ, but we believe that most air quality modelers are familiar with the terminology used in this paper. We respectfully request the reviewer to read the articles by Rao et al. (January 2011 issue of the *Bull. Amer. Meteor. Soc.*, pp 23-30), Dennis et al. (2010), Solazzo and Galmarini (2015), and Gilliam et al. (2015) included in the reference list.

---

## Referee Comment (RC2) · Anonymous Referee #2 · 5 Oct 2019

The paper presents some interesting findings about inherent uncertainties in chemistry transport model simulations of ozone concentrations. It is very well written, but sometimes hard to follow. In my opinion, some terms should be explained in more detail, before the paper can be published.

General comments:

The authors should explain why they think the data set they constructed by combining measured base line ozone with meteorology related short term variations from a 21 years CMAQ run could be seen as the output of a "perfect model". They claim that there is some inherent variability in the meteorological data that cannot be captured by any model system. However, reanalysis data may represent meteorologically related variations on time scales of few days very well, i.e. part of the variation included in the

SY component of the time series may be modelled quite well.

Why do you use 30+ years of ozone measurements for analyzing the observations while only 21 years can be used for comparisons to the model results? Wouldn't it be enough to look at the data set 1990 to 2010 for the observations, too? And why do you need to construct "pseudo ozone observations" and cannot use the observational data set as such? Please explain this in the text.

Specific comments:

Page 3, line 26: In equation (1) it should be made clear that the filter KZ(5,5) is applied to the ozone time series O3(t).

Page 4, line 17: Can you show by statistical evaluation that the SY component represents white noise.

Page 4, line 18: please explain AR(1)

Page 5, line 18-20: It isn't obvious for the reader which stations are at elevated sites.

Page 5, line 25: Define the "strength" of the SY component (being the standard deviation of the time series) here.

Page 6, line 8: Is there any reason why you selected this site?

Page 6, line 10/11: Is there an explanation why the model doesn't perform well for low concentrations?

Page 6, line 31/32: Couldn't this also be caused by emissions missing the correct temporal variation?

Page 7, line 9: How could these "slow changing processes" be improved in the models? What is the role of stratosphere/troposphere exchange which – to my knowledge – isn't well represented in the CMAQ model runs.

Page 7, line 13: See my comment above: I couldn't fully understand how you constructed the "perfect model data". As far as I understood it, they would in any case only be perfect for this model setup and model grid. Could you comment on this?

Caption of Figure 2: Explain the meaning of the number 420130801. Explain that the "observed" line in Fig 2a is for 2010.

Caption of Figure 3: Give the equations you refer to somewhere in this paper (e.g. in an appendix).

Figure 5: Use same colors for observations and model in both graphs.

Caption of Figure 6: "5)" should be "c)". "Light blue" in Fig 6d) appears to be grey. Figure 4 and Figure 7: Give units (ppb).

---

## Short Comment (SC1) · 7 Oct 2019

Rao et al. examine how the stochastic variability of the atmosphere affects the accuracy of regional air quality model predictions. Stochastic variability would be expected to introduce error in predictions even if the model is "perfect". This paper provides an analysis of the expected error. The question of the limits of "predictability" of models is well known in meteorology but it has not been explored very extensively for air quality models. Therefore, this paper provides a valuable contribution to the literature.

The paper provides an excellent basis for future research and improvements to air quality models as well. Atmospheric stochastic variability extends to scales that are well below current Eulerian model resolution. Eulerian models calculate gas-phase

chemical transformations across the modeling domain within grid-boxes and instanta-neous uniform mixing of chemical species is assumed for each grid-box. However, the stochastic variability of wind fields suggests that chemical concentrations should be represented by mean and varying components. This difference between reality and model representation may be most important for rapid, diffusion-limited reactions that affect ozone and particulate formation (Stockwell, J. of Meteor. and Atmos. Phys., 57, 159-172, 1995).
* * *

---

## Author Comment (AC2) · 26 Nov 2019

**Authors' Response (in red) to Referee #2's Comments and Changes in Manuscript (in blue)**

The paper presents some interesting findings about inherent uncertainties in chemistry transport model simulations of ozone concentrations. It is very well written, but sometimes hard to follow. In my opinion, some terms should be explained in more detail, before the paper can be published.

We thank the reviewer for the positive feedback on our paper.

General comments:

The authors should explain why they think the data set they constructed by combining measured base line ozone with meteorology related short term variations from a 21 years CMAQ run could be seen as the output of a "perfect model". They claim that there is some inherent variability in the meteorological data that cannot be captured by any model system. However, reanalysis data may represent meteorologically related variations on time scales of few days very well, i.e. part of the variation included in the SY component of the time series may be modelled quite well.

We have revised the discussion in Section 3.2 to clarify that this analysis combining the observed baseline component with 21 CMAQ synoptic components is meant to quantify the amount of model error present in the current simulations that could conceivably be reduced through improving the representation of synoptic-scale processes and/or increased horizontal resolution. As part of this revision, we no longer refer to this combination of the measured baseline and modeled synoptic component as "perfect model". Moreover, we have added text at the end of the introduction to clarify that when we refer to the errors that can be expected even from a "perfect" model with "perfect" inputs throughout the manuscript, we consider these errors to be those arising from atmospheric stochasticity which we estimate in Section 3.1 using historic observations. We also should have noted in our original manuscript that four-dimensional data assimilation (FDDA) was utilized following the methodology suggested by Gilliam et al. (see *Atmos. Environ.*, Vol.53, pp 186-201, 2012) and modified for fully-coupled meteorology-chemistry model applications as described in Hogrefe et al. (see *Atmos. Environ.*, Vol. 115, pp 683-694, 2015).  As the reviewer pointed out, the SY component in the model output contains some meteorologically related variations on time scales of few days.  The following revisions and additions have been made to address the reviewer's comment:

Revised and expanded the end of Section 1 (page 2, lines 29 – 32 in the original manuscript) as follows: "Also, no assessments have been made to date on the errors that are to be expected even from "perfect" regional-scale air quality modeling systems.  To estimate such irreducible model errors due to atmospheric stochasticity (which we consider to be the errors that are expected even from a "perfect" model with "perfect" inputs), we analyzed the observed daily maximum 8-hr (DM8HR) ozone time series data at monitoring locations across the contiguous United States (CONUS) during the 1981-2014 time period and present the results of this analysis in Section 3.1. In Section 3.2, we illustrate how this information could be used in guiding model development specifically aimed at addressing reducible errors in the synoptic component by contrasting the results from Section 3.1 with analysis using the

synoptic component from a 21-year simulation performed with the fully coupled WRF-CMAQ simulations covering the 1990-2010 period."

Expanded the model description in Section 2 (added after page 3 line 7 in the original manuscript) as follows: "To ensure better characterization of the prevailing meteorology (i.e., synoptic forcing) in the retrospective 21-year WRF-CMAQ simulations, four-dimensional data assimilation (FDDA) was utilized following the methodology suggested by Gilliam et al. (2012) and modified for fully-coupled meteorology-chemistry model applications as described in Hogrefe et al. (2015). The model set-up and performance evaluation of these historical multiyear WRF-CMAQ simulations have been published by Xing et al. (2015), Gan et al. (2015), and Astitha et al. (2017)."

Modified the discussion in Section 3.2 (page 5, line 23 – page 6, line 9 in the original manuscript) as follows: "In this section, we analyze long-term records of model simulations in an attempt to quantify the error associated with the modeled SY component that results both from not explicitly representing stochastic variations in atmospheric dynamics and emission variability in the current generation regional air quality models and from other reducible sources of model error. … To isolate the impact of model imperfections on only the SY time scale on errors across the ozone distribution, we assume that the model perfectly reproduces the 'true' BL depicted by the observed 2010 BL. We then use this 'perfect' modeled BL and reconstruct 'pseudo-simulated' ozone time series, similar to what was done in Fig. 3, except for using the SY component embedded in the 21 years of coupled WRF-CMAQ simulations. The rationale for this analysis is to quantify the amount of model error present in the current simulations that could conceivably be reduced through improving the representation of synoptic and mesoscale processes and/or increased horizontal resolution with appropriate data assimilation techniques."

Why do you use 30+ years of ozone measurements for analyzing the observations while only 21 years can be used for comparisons to the model results? Wouldn't it be enough to look at the data set 1990 to 2010 for the observations, too? And why do you need to construct "pseudo ozone observations" and cannot use the observational data set as such? Please explain this in the text.

As noted in our paper, any observation at a given percentile represents an event or a single realization out of a population. The object of our paper is to quantify the inherent variability in the observations due to the stochastic nature of the atmosphere. To this end, we thought that the use of 30+ years of historical data rather than 21 years would help in making more robust estimates of the expected errors even from "perfect" models driven with "perfect" input. As demonstrated in our previous research (see Porter, et al., 2017 *Atm. Poll. Res.*; Astitha, et al., 2017 and Luo, et al., 2019 in *Atm. Env.*), the baseline forcing can be viewed as the deterministic part in observations while the SY forcing is the near-stochastic part. We superimposed 30+ adjusted SY forcing on the baseline embedded in 2010 ozone observations to generate 30+ representations of ozone concentrations, which are equally likely to occur at any given probability point stemming from the stochastic nature of the atmosphere. Details on this approach can be found in Luo, et al. (2019 AE). The discussion in Section 3.1 of the revised manuscript was modified in response to the reviewer's comments.

Updated the second paragraph of Section 3.1 (page 4, line 25 – page 5, line 11 in the original manuscript) as follows: "Once the scale separation is achieved with the KZ5,5, we superimposed the SY forcing imbedded in 30+ years of historical DM8HR ozone time series measured at a given location on the baseline component of the ozone time series at that location to generate 30+ reconstructed or pseudo ozone distributions. Illustrative results using eq. (3) at a suburban location in Altoona, PA are presented for 2010 base year in Fig. 3a … note, it is equally likely for any of these 30+ CDFs to occur because of the stochastic nature of the atmosphere even though the individual event in 2010 yielded the CDF shown in red. As mentioned before, ozone mixing ratio at any given probability point on the red line in Fig. 3a reflects an individual event while ozone values at the same probability in different CDFs (gray lines) reflect the population stemming from the stochastic nature of the atmosphere. In other words, there are 30+ dynamically consistent ozone time series attributable to the 2010 baseline (given 2010 emissions) for examining the inherent variability due to atmospheric stochasticity. … Using these 30+ pseudo-observation ozone mixing ratios and the actual observed ozone values at each percentile, statistical metrics such as Bias, RMSE, coefficient of variation (CV=standard deviation/mean), normalized mean error (NME) and normalized mean bias (NMB) are presented in Fig. 3b and c (see Emery et al. (2016) for the description of the statistical metrics considered here). … The extreme values are better described in statistical terms rather than in deterministic sense (Hogrefe and Rao, 2001)."

Specific comments:

Page 3, line 26: In equation (1) it should be made clear that the filter KZ(5,5) is applied to the ozone time series O3(t).

Yes, the reviewer is correct that the SY component is estimated by applying the KZ filter with a window length of 5 days and 5 iterations to the ozone time series O3(t) as described in Porter et al. (2015), Rao et al. (2011), and Luo et al. (2019). We updated the notation in equations (1) and (2) and also changed occurrences of KZ(5,5) in the text to $KZ_{5,5}$ to better reflect the operation of the filter.

Equations 1 and 2: changed "KZ(5,5)" to "$KZ_{5,5}(O_3(t))$". Also changed all occurrences of KZ(5,5) in the text to $KZ_{5,5}$

Page 4, line 17: Can you show by statistical evaluation that the SY component represents white noise.

Because we haven't used the concept of white noise process to statistically model the SY forcing in our paper, we've removed the reference to white noise in the revised manuscript. However, to respond to the referee's question, we display below the autocorrelation function for time series in SY forcing and white noise. It is evident that the correlation drops off after 1-day lag.

[Figure]

In addition, we display below the autocovariance function as a function of lag (days), extracted from the article titled "Dealing with the ozone non-attainment problem in the Eastern United States" by Rao et al. on Page 20 in the January 1996 issue of the EM Magazine, a publication of the Air & Waste Management Association. These results reveal that the short-term variation (SY component) in observed ozone time series data is statistically indistinguishable from "white noise" with an autocorrelation coefficient that drops to zero at a lag of 1 day.

[Figure]

Page 4, line 18: please explain AR(1)

An AR(1) autoregressive process is the first-order process, meaning that the current value is based on the immediately preceding value. However, since we removed the references to "white noise" and AR(1) from our manuscript, this explanation has not been added to the revised manuscript.

Page 5, line 18-20: It isn't obvious for the reader which stations are at elevated sites.

We agree that adding this information would be useful for the reader and have added a panel to Figure 5 that shows the elevation of each monitoring site over CONUS.

The following new figure (Figure 5c) has been added to the revised manuscript:

[Figure]

Page 5, line 25: Define the "strength" of the SY component (being the standard deviation of the time series) here.

The definition has been added to the revised manuscript as follows:

"it should be noted that the linear relationship between the strength of SY (defined as the standard deviation of the data in the synoptic component) and the magnitude of the BL (defined as the mean of the data in the baseline component) has been taken into account in generating 30+ years of adjusted SY forcing as illustrated in Luo et al. (2019)"

Page 6, line 8: Is there any reason why you selected this site?

It has complete data for 30+ years.  We could have picked another site; they all exhibit same features.

Page 6, line 10/11: Is there an explanation why the model doesn't perform well for low concentrations?

The CMAQ team at the U.S. Environmental Protection Agency is investigating the reasons for model's poor performance at the low end of the concentration distribution.  Plausible causes include 36-km grid spacing not resolving the effects of NO titration in urban areas, errors in atmospheric deposition, representation of variability in background concentrations of $O_3$, precursor and reservoir species, etc. Also, it should be noted that the stochastic variability affects mostly the lower and upper tails of the pollutant concentration distribution.

Page 6, line 31/32: Couldn't this also be caused by emissions missing the correct temporal variation?

Yes, it is possible. To address the reviewer's comment, we have expanded an earlier sentence that also discusses the limitations of the current model setup as follows:

Revised sentence (page 6, lines 1-2 in the original manuscript): "The 36-km grid may be better representing the large-scale synoptic forcing associated with the translation of weather systems than the meso-scale weather and urban influences (both dynamics and emission driven) that are embedded in the observed SY component."

Page 7, line 9: How could these "slow changing processes" be improved in the models? What is the role of stratosphere/troposphere exchange which – to my knowledge – isn't well represented in the CMAQ model runs.

As already indicated in the manuscript discussion, one needs to pay more attention to properly specifying the lateral boundary conditions, duration/strength of stratosphere-troposphere exchanges, Madden-Julian Oscillation (MJO), ENSO, climate change, control policies, spatio-temporal variability in emissions loading, etc. The hemispheric CMAQ simulations used to drive the regional CMAQ runs used here employed potential vorticity-based scaling to represent $O_3$ in the model's upper troposphere-lower stratosphere (UTLS). The method was subsequently enhanced to represent seasonal and latitudinal dependencies in the relationship between potential vorticity and ozone and improved the 3-dimensional $O_3$ distribution represented by the model as well as the seasonal impacts of STE on lower tropospheric and surface-level $O_3$ as detailed in analyses presented in Xing et al. (2016) and Mathur et al. (2017). Interested readers are pointed to these studies and references are also included in the discussion.

Modified the second paragraph of Section 2 (Page 3, Lines 11-19 in the original manuscript) and added two references as follows: "It has been shown that time series of the daily maximum 8-hr ozone concentrations contain fluctuations operating on different time scales (e.g., intra-day forcing induced by the fast-changing emissions and atmospheric boundary layer evolution; diurnal forcing induced by the day and night differences; synoptic forcing induced by the passage of weather systems across the country, sub-seasonal forcing due to Madden-Julian Oscillation (MJO), and long-term forcing induced by emissions, El-Nino-Southern Oscillation (ENSO), climate change, and other slow-varying processes such as seasonal and sub-seasonal variations in the atmospheric deposition and stratosphere-troposphere exchange processes) as noted by Rao et al. (1997), Vukovich, (1997),  Hogrefe et al. (2000),  Porter et al. (2015), Astitha et al. (2017), Xing et al. (2016), and Mathur et al. (2017))."

Mathur, R.; Xing, J.; Gilliam, R.; Sarwar, G.; Hogrefe, C.; Pleim, J.; Pouliot, G.; Roselle, S.; Spero, T.L.; Wong, D.C.; Young, J. Extending the Community Multiscale Air Quality (CMAQ) modeling system to hemispheric scales: overview of process considerations and initial applications. *Atmos. Chem. Phys*. 2017, 17, 12449-12474, https://doi.org/10.5194/acp-17-12449-2017.

Xing, J., R. Mathur, J. Pleim, C. Hogrefe, J. Wang, C.-M. Gan, G. Sarwar, D. Wong, and S. McKeen, Representing the effects of stratosphere-troposphere exchange on 3D $O_3$ distributions in chemistry transport models using a potential vorticity based parameterization, Atmos. Chem. Phys., 16, 10865-10877, doi:10.5194/acp-16-10865-2016, 2016

Page 7, line 13: See my comment above: I couldn't fully understand how you constructed the "perfect model data". As far as I understood it, they would in any case only be perfect for this model setup and model grid. Could you comment on this?

We have revised the discussion in Section 3.2 to clarify that this analysis combining the observed baseline component with 21 CMAQ synoptic components is meant to quantify the amount of model error present in the current simulations that could conceivably be reduced through improving the representation of synoptic-scale processes and/or increased horizontal resolution. As part of this revision, we no longer refer to this combination of the measured baseline and modeled synoptic component as "perfect model". Moreover, we have added text at the end of the introduction to clarify that when we refer to the errors that can be expected even from a "perfect" model with "perfect" inputs throughout the manuscript, we consider these errors to be those arising from atmospheric stochasticity which we estimate in Section 3.1 using historic observations. We also should have noted in our original manuscript that four-dimensional data assimilation (FDDA) was utilized following the methodology suggested by Gilliam et al. (see *Atmos. Environ*., Vol.53, pp 186-201, 2012) and modified for fully-coupled meteorology-chemistry model applications as described in Hogrefe et al. (see *Atmos. Environ.*, Vol. 115, pp 683-694, 2015).  As the reviewer pointed out, the SY component in the model output contains some meteorologically related variations on time scales of few days.  The following revisions and additions have been made to address the reviewer's comment:

Revised and expanded the end of Section 1 (page 2, lines 29 – 32 in the original manuscript) as follows: "Also, no assessments have been made to date on the errors that are to be expected even from "perfect" regional-scale air quality modeling systems.  To estimate such irreducible model errors due to atmospheric stochasticity (which we consider to be the errors that are expected even from a "perfect" model with "perfect" inputs), we analyzed the observed daily maximum 8-hr (DM8HR) ozone time series data at monitoring locations across the contiguous United States (CONUS) during the 1981-2014 time period and present the results of this analysis in Section 3.1. In Section 3.2, we illustrate how this information could be used in guiding model development specifically aimed at addressing reducible errors in the synoptic component by contrasting the results from Section 3.1 with analysis using the synoptic component from a 21-year simulation performed with the fully coupled WRF-CMAQ simulations covering the 1990-2010 period."

Expanded the model description in Section 2 (added after page 3 line 7 in the original manuscript) as follows:  "To ensure better characterization of the prevailing meteorology (i.e., synoptic forcing) in the retrospective 21-year WRF-CMAQ simulations, four-dimensional data assimilation (FDDA) was utilized following the methodology suggested by Gilliam et al. (2012) and modified for fully-coupled meteorology-chemistry model applications as described in Hogrefe et al. (2015).  The model set-up and performance evaluation of these historical multiyear WRF-CMAQ simulations have been published by Xing et al. (2015), Gan et al. (2015), and Astitha et al. (2017)."

Modified the discussion in Section 3.2 (page 5, line 23 – page 6, line 9 in the original manuscript) as follows: "In this section, we analyze long-term records of model simulations in an attempt to quantify the error associated with the modeled SY component that results both from not explicitly representing stochastic variations in atmospheric dynamics and emission variability in the current generation regional air quality models and from other reducible sources of model error. … To isolate the impact of model imperfections on only the SY time scale on errors across the ozone distribution, we assume that the model perfectly reproduces the 'true' BL depicted by the observed 2010 BL.  We then use this 'perfect' modeled BL and reconstruct 'pseudo-simulated' ozone time series, similar to what was done in Fig. 3, except for using the SY component embedded in the 21 years of coupled WRF-CMAQ simulations. The rationale for this analysis is to quantify the amount of model error present in the current simulations that could conceivably be reduced through improving the representation of synoptic and mesoscale processes and/or increased horizontal resolution with appropriate data assimilation techniques."

Caption of Figure 2: Explain the meaning of the number 420130801. Explain that the "observed" line in Fig 2a is for 2010.

The number represents the site # in EPA's AQS database; it has no specific meaning other than being an identifier for the location of the monitoring site.

The figure caption (Figure 2 in the original manuscript, Figure 3 in the revised manuscript) has been updated as follows: "Figure 3a: Comparison between the observed cumulative distribution function (CDF) for 2010 shown in red with 30+ pseudo-observations CDFs generated from historical DM8HR ozone time series shown in gray at a suburban site at Altoona in PA (AQS station identifier 420130801). The blue line represents the average of the 30+ gray lines; Figure 3b: Display of various statistical metrics (standard deviation (std), root mean square error (RMSE), bias) derived by comparing the actual observed and pseudo ozone values in Fig. 3a; Figure 3c: Normalized statistical metrics of normalized mean error (NME), normalized mean bias (NMB), coefficient of variation (CV). Notice the large variability occurring at the lower and upper percentiles"

Caption of Figure 3: Give the equations you refer to somewhere in this paper (e.g. in an appendix).

The paper by Emery et al. (2016) and many other papers included in the references list recommended the statistical metrics such as RMSE, NME, Bias, NMB, CV, etc. whose definitions are well known.

Updated the second paragraph of Section 3.1 (page 5 line 9 of the original manuscript) in which Figure 3 (Figure 2 in the original manuscript) is discussed as follows: "see Emery et al. (2016) for the description of the statistical metrics considered here"

Figure 5: Use same colors for observations and model in both graphs.

Done.

Caption of Figure 6: "5)" should be "c)". "Light blue" in Fig 6d) appears to be grey. Figure 4 and Figure 7: Give units (ppb).

Done.

---

## Author Comment (AC3) · 26 Nov 2019

Authors' Response (in red) to Dr. William Stockwell's Comments

Rao et al. examine how the stochastic variability of the atmosphere affects the accuracy of regional air quality model predictions. Stochastic variability would be expected to introduce error in predictions even if the model is "perfect". This paper provides an analysis of the expected error. The question of the limits of "predictability" of models is well known in meteorology but it has not been explored very extensively for air quality models. Therefore, this paper provides a valuable contribution to the literature. The paper provides an excellent basis for future research and improvements to air quality models as well.

We thank Dr. Stockwell for his positive feedback on our paper.

Atmospheric stochastic variability extends to scales that are well below current Eulerian model resolution. Eulerian models calculate gas-phase chemical transformations across the modeling domain within grid-boxes and instantaneous uniform mixing of chemical species is assumed for each grid-box. However, the stochastic variability of wind fields suggests that chemical concentrations should be represented by mean and varying components. This difference between reality and model representation may be most important for rapid, diffusion-limited reactions that affect ozone and particulate formation (Stockwell, J. of Meteor. and Atmos. Phys., 57, 159-172, 1995).

Agreed.  We hope that the next generation of operational Eulerian models would be able to handle the physical and chemical processes as suggested by Dr. Stockwell.  Also, it is important to resolve emissions inventories to the time and space scales of the model.

---

## Author Comment (AC4) · 26 Nov 2019

**Authors' Responses (in red) to the Comments on acp-2019-642 by Anonymous Referee # 1 and Changes Made (in blue) in the Manuscript**

This manuscript separated short-term synoptic-scale fluctuations from long-term baseline component embedded in the daily maximum 8-hr ozone time series using a filter and estimated the limit of air quality model's accuracy (or predictability/uncertainties of air quality prediction). This is an interesting topic for air quality prediction.

The authors thank the referee for recognizing the importance of our work.

But to my surprise, the authors did not even consider lead time when discussing air quality predictability (or limit of air quality prediction). What is the configuration of the air quality prediction? Was this one-day prediction? Two-day prediction? Prediction uncertainties/errors will change significantly with different lead time.

It seems that the referee has misunderstood the modeling simulations we have examined in this paper. Please note that our paper focused on the evaluation of retrospective simulations of 21-years (1990 to 2000) of the daily maximum 8-hour average ozone concentrations over the contiguous United States (CONUS) with the fully-coupled Weather Research Forecasting (WRF) meteorological model and the Community Multiscale Air Quality (CMAQ) chemical transport model.  While the concepts presented in our paper are applicable to examining air quality forecasting products, the results of our study reflect the prediction capability of the model based on retrospective simulations and **not** air quality forecasting. To ensure better characterization of the prevailing meteorology (i.e., synoptic forcing) for these retrospective 21-year simulations, four-dimensional data assimilation (FDDA) was utilized following the methodology suggested by Gilliam et al. (see *Atmos. Environ*., Vol.53, pp 186-201, 2012) and modified for fully-coupled meteorology-chemistry model applications as described in Hogrefe et al. (see *Atmos. Environ.*, Vol. 115, pp 683-694, 2015).  The modeling set-up and performance evaluation of these historical multiyear WRF-CMAQ simulations have been published by Xing et al. (2015), Gan et al. (2015), and Astitha et al. (2017) as referenced in our manuscript.  The following material has been added in the revised manuscript.

Expanded the model description in Section 2 (added after page 3 line 7 in the original manuscript) as follows:  "To ensure better characterization of the prevailing meteorology (i.e., synoptic forcing) in the retrospective 21-year WRF-CMAQ simulations, four-dimensional data assimilation (FDDA) was utilized following the methodology suggested by Gilliam et al. (2012) and modified for fully-coupled meteorology-chemistry model applications as described in Hogrefe et al. (2015).  The model set-up and performance evaluation of these historical multiyear WRF-CMAQ simulations have been published by Xing et al. (2015), Gan et al. (2015), and Astitha et al. (2017)."

It is also surprising to see the authors suggesting improving simulation of the baseline concentration by focusing on the quality of the emission inventory and the model's treatment for the slow-changing atmospheric processes. I have no question for improving emission inventory, but I am confused by improving the slow-changing atmospheric processes.

A number of papers have been published, documenting the importance of the baseline (longer-term) forcing embedded in ambient ozone data (see e.g., Rao et al., 1996 and 1997; Hogrefe et al., 2000; Rao et al., 2011; Porter et al., 2017; Astitha et al., 2017; Luo et al., 2019).  As noted in Astitha et al. (2017), the baseline level and the strength of the synoptic forcing are to be viewed as the necessary and sufficient conditions for observing peak ozone levels.  Rao et al. (2011), Hogrefe et al. (2000), Astitha et al. (2017), Porter et al (2017), and Luo et al. (2019) have demonstrated that when the magnitude of the baseline level is low, there will not be ozone exceedances of the USA's National Ambient Air Quality Standard no matter how strong the synoptic forcing is.  In this study, we are working with the model (WRF-CMAQ)-predicted and corresponding observed time series of the daily maximum 8-hour ozone concentrations during 1990 to 2000 at various monitoring locations over CONUS; therefore, the Nyquist interval here is 2-days.  Using both Empirical Mode Decomposition (EMD) and KZ filtering, we separated the synoptic forcing (time scale < 24 days) and baseline (time scale > 1 month) forcing embedded in the time series of observed and modeled daily maximum 8-hour ozone concentrations.  To illustrate, the results of the application of EMD to the daily maximum 8-hr ozone time series data measured at Altoona, PA are presented in Fig. 1 below.  The top left panel displays the raw ozone time series while the top of the right panel shows its power spectrum.  The 7 intrinsic mode functions (IMFs) and the residual on the left side, and their corresponding power spectra on the right reveal that most of the synoptic-scale features in ozone data are reflected in IMFs 1 and 2.  The baseline ozone is extracted by removing the first two IMFs from the raw ozone time series.

[Figure]

**Figure 1.  Results of the application of the EMD technique, which is designed for analyzing non-stationary and non-linear time series data, to the daily maximum 8-hour ozone time series data at the Altoona, PA site.  The numbers on the right side represent the time scale (in days) associated with each IMF.  Note, the power spectrum of raw ozone time series shows that the energy in the 1-10 days (SY forcing) is an order of magnitude less than that in the longer (baseline) time scale.**

The scale separation achieved from the application of EMD and KZ filter, displayed in Fig. 2, reveal that the results are quite similar.  It should be noted that the short-term component (SY) extracted from the KZ filter as well as EMD's high-frequency IMFs 1 and 2 resemble white noise process.

[Figure]

**Figure 2.  The top left panel displays the raw daily maximum 8-hour ozone time series together with the baselines extracted from the KZ filter and EMD while the top right panel reveals the similarity between white noise and synoptic forcing imbedded in these observed ozone time series.  The bottom two panels compare the power spectra of the baseline forcing (left) and the synoptic forcing (right) derived from KZ filtering and EMD (sum of IMF 1 and IMF2).  Notice that most of the energy in the baseline time series is in the longer time scale while most of the energy of the short-term component is in the high-frequency range.  The similarity of results from both scale separation techniques demonstrates that the two scales of interest (i.e., baseline and synoptic forcing) have been extracted reasonably well.**

This material to demonstrate good scale separation has been added in the revised manuscript.

Added new material to the beginning of Section 3.1 (page 4 line 15 in the original manuscript): "Using both Improved CEEMDAN and KZ filtering, we separated the synoptic forcing (time scale < 24 days) and baseline (time scale > 1 month) forcing embedded in the time series of observed and modeled daily maximum 8-hour ozone concentrations.  To illustrate, the results from the application of Improved CEEMDAN to the daily maximum 8-hr ozone time series data measured at Altoona, PA are presented in

Fig. 1. The top left panel displays the raw ozone time series while the top of the right panel shows its power spectrum. The 7 intrinsic mode functions (IMFs) and the residual on the left side, and their corresponding power spectra on the right reveal that most of the synoptic-scale features in ozone data are imbedded in IMFs 1 and 2.  The baseline ozone is extracted by removing the first two IMFs from the raw ozone time series. To illustrate the concept of the ozone baseline, DM8HR time series measured in 2010 at Altoona, PA is presented in Fig. 2a together with the embedded baseline concentration as extracted by the KZ5,5 and Improved CEEMDAN. It is evident that high ozone levels are always associated with the elevated baseline. The difference between the raw ozone time series and baseline, denoted as the short-term or synoptic forcing (SY), is displayed in Fig. 2b.  The power spectra, displayed in Figs. 2c and d, reveal both methods yielded good scale separation. Due to the good agreement between both scale separation techniques, only the results from the KZ filter are presented for the remainder of the manuscript."

The huge advances of weather prediction during the past few decades has been focusing on 1-day or 2-day prediction. On such short-term synoptic-scale weather processes, our weather prediction did excellent job and has been improved through years. Such improvement can benefit air quality prediction significantly (Zhang et al., 2007). I would thus imagine that such short-term practical predictability of air quality can be much improved through better model treatments and better initial conditions of meteorological and chemical variables, as well as emissions.

As stated before, our study deals with evaluating the retrospective air quality simulations, not modeling results from an air quality forecasting effort.  Air quality modeling uncertainty even for the retrospective modeling cases, outside of the chemistry formulation and boundary conditions, is attributed primarily to meteorology and emissions inputs.  Vautard et al. (*Atm. Environ*., Vol. 53, pp 15-37, 2012) concluded that major challenges still remain in the simulation of prevailing meteorology (e.g., errors in wind speed, PBL, nighttime meteorology, clouds) in retrospective air quality modeling.  Based on retrospective ozone episodic modeling with the WRF-CMAQ model using various sets of equally likely initial conditions for meteorology along with FDDA, Gilliam et al. (2015) confirmed the presence of sizable spread in WRF solutions, including common weather variables of temperature, wind, boundary layer depth, clouds, and radiation, thereby causing a relatively large range of ozone concentrations.  Also, pollutant transport is altered by hundreds of kilometers over several days.  Ozone concentrations of the ensemble varied as much as 10–20 ppb (or 20–30%) in areas that typically have higher pollution levels.

We acknowledge that air quality forecasting often is concerned with accurately predicting transient pollution events. The analysis of modeling uncertainty in our retrospective simulations that employed data assimilation suggest that the largest improvement in forecast accuracy can be achieved through implementing bias correction techniques (i.e. by correcting for systematic errors such as those caused by emission uncertainties that manifest themselves primarily in the baseline) as demonstrated by Kang et al. (2008 *JGR-Atmospheres*, Vol. 113, Issue D23308; 2010 *Atmos. Environ*., Vol. 44, 18, pp 2203-2212). However, we do not dispute the fact that any advances in predicting short-term atmospheric processes or phenomena, be it through model improvements, better initial conditions, or ensemble techniques, may help to reduce the portion of the prediction error that is not due to systematic biases. Rather, we argue through our analysis of both observations and retrospective air quality simulations that, at least

for retrospective air quality planning applications, the focus of model development and evaluation efforts should be on longer time scales. The following discussion has been included in the revised manuscript.

Expanded the relevant sentences in the conclusions (page 7, lines 9 – 11 of the original manuscript) as follows: "To improve regional-scale ozone air quality models, attention should be paid to accurately simulate the baseline concentration by focusing on the quality of the emission inventory and the model's treatment for the boundary conditions and slow-changing (operating on sub-seasonal, seasonal, and longer-term time scales) atmospheric processes. Also, errors in reproducing the synoptic forcing can possibly be reduced with high-resolution meteorological modeling using appropriate data assimilation techniques."

Other specific comments: Many of the concept/discussion regarding inherent/practical predictability, reducible/irreducible uncertainties are questionable/wrong, or different from those used in weather prediction. For example, emissions are definitely reducible uncertainties and factors in practical predictability. Please carefully define those terms/concepts and refer to normally used/accepted definitions. The writing is overly concise, particularly in many cases where detailed explanation is needed. References: Zhang, F., Bei, N., Nielsen-Gammon, J. W., Li, G., Zhang, R., Stuart, A., & Aksoy, A. (2007). Impacts of meteorological uncertainties on ozone pollution predictability estimated through meteorological and photochemical ensemble forecasts. Journal of Geophysical Research, 112, D04304. https://doi.org/10.1029/2006JD007429 Interactive comment on Atmos. Chem. Phys. Discuss., https://doi.org/10.5194/acp-2019-642, 2019.

It is difficult to identify exactly which terms the reviewer is referring to in comment that terms are not defined or are not defined as "normally used/accepted".  From our perspective, we are consistent with the reviewer's example (we agree that uncertainty in retrospectively simulated ozone concentrations can be improved with improvement in emissions characterizations).  The confusion in the definition of terms may be more to do with the reviewer's misunderstanding that the manuscript was focused on air quality forecasting rather than on analysis of retrospective simulations.   We agree that terminologies between the weather forecasting community and the air quality modeling community may differ, but we believe that most air quality modelers are familiar with the terminology used in this paper.  We respectfully request the reviewer to read the articles by Rao et al. (January 2011 issue of the *Bull. Amer. Meteor. Soc.*, pp 23-30), Dennis et al. (2010), Solazzo and Galmarini (2015), and Gilliam et al. (2015) included in the reference list.

---

## Author Response (AR2)

**Authors' Responses (in red) to the Comments on the Revised Version of acp-2019-642 by Anonymous Referee # 1 on 16 Dec 2019 and Changes Made (in blue) in the Manuscript**

Thanks for clarifying that this study only applies to retrospective simulations rather than real forecasting. This is very very important! Please explicitly mention this in the article title and abstract

We thank the referee for taking the time to review our responses to the previous comments and the revised manuscript. Please note that the main object of our paper is to identify the expected errors in regional air quality models since the current generation of models do not explicitly treat the stochastic nature of the atmosphere. To this end, we have analyzed multi-decadal historical ozone observations from several locations over the contiguous United States and estimated the expected root mean square error (RMSE) from current operational deterministic models driven with "perfect" input. Because ours is an observations-based methodology, the results presented here are applicable to both forecasting and retrospective simulations from deterministic models. Therefore, we see no need to change the title for our article. However, we modified the abstract and text to clarify the types of errors that we are dealing with in this paper as referee suggested.

Again, many of the concept/discussion regarding inherent/practical predictability, reducible/irreducible uncertainties are wrong.

For example, emissions are definitely reducible uncertainties and factors in practical predictability (10.1175/JAS-D-17-0157.1 10.1029/2018MS001457) while it is categorised as inherent or irreducible uncertainties/predictability in this manuscript.

We agree that emissions are comprised mostly of reducible (systematic) errors, but our paper does not deal with the errors attributable to the uncertainties in the emission inventories. In this paper, we address only the unsystematic or random errors stemming from the stochastic nature of the atmosphere. Also, we never used the term "predictability" in our paper.

See some definitions:

In Ying and Zhang (2018): Intrinsic predictability refers to the ability to predict given nearly perfect representation of the dynamical system (by a forecast model) and nearly perfect inputs.

Emissions are inputs for air quality forecast models. Apparently emissions are not nearly perfect in this study (or in any studies), thus prediction uncertainties due to emission uncertainties are not intrinsic predictability.

In Thomas et al. (2019), it is explicitly mentioned that emissions pertains to practical predictability rather than inherent predictability.

Please carefully define those terms/concepts and refer to normally used/accepted definitions.

We have clarified the definitions so the air quality modeling community can better understand the problem addressed in this article (namely, stochasticity). We have referred to the 2 articles suggested by the referee in this revision. Some researchers claim that ensemble modeling will help address prediction uncertainties, but the range of values predicted from different modeling systems reflects the structural and parametric uncertainty, which are reducible errors because the model structure and its formulation can continually be improved with as new knowledge emerges (e.g., Solazzo et al., https://doi.org/10.1016/j.atmosenv.2012.01.003; Galmarini et al., https://doi.org/10.5194/acp-18-8727-2018; Solazzo et al., https://doi.org/10.5194/acp-17-3001-2017, 2017).

Our team has published numerous papers on model sensitivities attributable to the meteorology (e.g., see Biswas and Rao, https://doi.org/10.1175/1520-0450(2001)040<0117:UIEOMS>2.0.CO;2; Biswas, et al., 11th Joint Conference on the Applications of Air Pollution Meteorology with the Air & Waste Management Association; Zhang et al., https://doi.org/10.1023/A:1011557402158; Sistla et al., https://doi.org/10.1007/BF00480816; Gego et al., https://doi.org/10.1007/s10652-005-0486-3; Wolke et al., https://doi.org/10.1016/j.atmosenv.2012.02.085; Makar et al., https://doi.org/10.1016/j.atmosenv.2014.10.021; Watson et al., https://doi.org/10.1016/j.atmosenv.2015.07.037; Kong et al., https://doi.org/10.1016/j.atmosenv.2014.09.020), attributable to emissions (e.g., Rao et al., EPA-230/2-89-026; Ku et al., https://doi.org/10.1023/A:1011513603066;, Rao and Sistla, https://doi.org/10.1007/BF00480816; Im et al., https://doi.org/10.5194/acp-18-8929-2018; Solazzo et al., https://doi.org/10.5194/acp-17-10435-2017, 2017; Napelenok et al., https://doi.org/10.1016/j.atmosenv.2011.03.030; Pierce et al., https://doi.org/10.1016/j.atmosenv.2010.05.046), and attributable to chemical mechanism (e.g., Sarwar et al., https://doi.org/10.1016/j.atmosenv.2019.06.020; Sarwar et al., https://doi.org/10.1016/j.atmosenv.2007.12.065; Sarwar et al., https://doi.org/10.1016/j.atmosenv.2009.09.012; Kitayama et al., https://doi.org/10.1016/j.atmosenv.2018.11.003). Some of the papers relevant to our article are included in the refences.

It should be noted that there will always be differences between a model estimate and it's corresponding observation because (1) what's predicted isn't what's observed; that is, while an observation at any given time and location represents a single event out of a population, the predicted value reflects the population average, and (2) the problem of point measurement vs. volume-averaged concentration. This paper deals with only the first point since it is linked to

the stochastic nature of the atmosphere. We hope that the next generation of the operational model will be a "regional-scale deterministic-stochastic modeling system" that can explicitly treat the mean and fluctuation components for all variables simultaneously in simulating air quality. Until such deterministic-stochastic models become available, the atmospheric stochasticity imposes a limit to the accuracy that can be expected from the regional-scale air quality models currently being used for forecasting and regulatory purposes. This paper quantifies the errors that should be expected for various percentiles of the concentration cumulative distribution function.

Overview of changed sections in the manuscript (see tracked changes version for details):

[revised manuscript text omitted]